# DIFFERENTIALLY PRIVATE PRINCIPAL COMPONENT ANALYSIS FOR VERTICALLY PARTITIONED DATA

## ABSTRACT

We study the problem of differentially private principal component analysis (DP PCA) for vertically partitioned data. In this setting, an untrusted server wants to learn the optimal rank-$k$ subspace of an underlying sensitive dataset $D$, which is partitioned among multiple clients by attributes/columns. While differential privacy has been heavily studied for horizontally partitioned data (namely, when $D$ is partitioned among clients by records/rows), its applications on vertically partitioned data are very limited. To fill this gap, we propose SPCA, which introduces noise of scale comparable with the strong centralized baseline to the obtained subspace while preserving DP without assuming any trusted client or third party. The theoretical analysis shows that our solution is able to match the privacy-utility trade-off of the optimal baseline in the centralized setting. Finally, we provide experiments on real-world datasets to validate the theoretical analysis.

## 1 INTRODUCTION

Data privacy is a critical concern in federated learning (FL) (McMahan et al., 2016), where an untrusted server and multiple clients train a model collaboratively on distributed data. Secure multiparty computation (MPC) (Yao, 1986; Chaum et al., 1987) has been applied to FL to enhance collaboration while preserving privacy. With MPC, clients are able to *collaboratively* compute a function (e.g., the model update) without exposing their sensitive input local data (e.g., see Bonawitz et al. (2017); Mohassel & Zhang (2017)). Although MPC solves the problem of *how* to compute the outcome, *what* MPC computes (namely, the outcome itself) still leaks information about the inputs (e.g., see Hitaj et al. (2017); Melis et al. (2019)).

Differential privacy (DP) (Dwork et al., 2006) is widely adopted in *horizontal* federated learning (HFL) to preserve data privacy, where the sensitive data is partitioned among clients according to records/rows, a common scenario for the cross-device setting. The idea of DP is to let each client independently perturb the information (e.g., the model update) computed on her local partition before sharing it with the untrusted server or other clients. An adversary who sees the perturbed *outcome* cannot infer the real data with high confidence (McMahan et al., 2018; Kairouz et al., 2021). Hence, privacy is preserved. However, only a limited selection of works, such as Wu et al. (2020); Xu et al. (2021); Ranbaduge & Ding (2022), study privacy in *vertical* federated learning (VFL), where the sensitive data is partitioned by clients according to attributes/columns, a common scenario for the cross-silo setting.

In this paper, we study the problem of principal component analysis (PCA) (Pearson, 1901; Jolliffe, 2014) under vertical federated learning. PCA aims at reducing the number of attributes of a given dataset while capturing its original information. PCA has been widely applied to areas such as image processing, biostatistics, and finance (e.g., see Novembre & Stephens (2008); Abadi et al. (2016b)). In our setting, a sensitive dataset $D$ is partitioned by multiple clients (who do not trust each other) according to attributes/columns. An untrusted server wants to obtain a subspace learned from the dataset such that the variance of the original dataset is mostly preserved on the subspace (referred to as the principal components). In terms of privacy, we aim to protect every individual record from the server, as well as the local partition of every participating client from other clients in FL.

**Related works.** We first briefly explain why existing solutions for DP in HFL do not apply to our setting. The key is *non-linearity*. In HFL, the untrusted server is often interested in some linear function (e.g., the sum/mean of local model updates), where each component can be computed

independently by a single client on her local data alone. As a result, each client is able to perturb her local component (e.g., local model update) before the aggregation with other clients, which produces a global estimate released to the server (Kairouz et al., 2021). The computation for PCA over vertically partitioned data, on the other hand, is not linear and requires *different clients* to participate at the same time (as we will see in Section 3.3). Hence, the existing DP algorithms from HFL do not work in VFL. Due to the same reason, the existing solutions for DP PCA in HFL, such as Wang & Chang (2018); Wang & Xu (2020); Gu et al. (2022), do not apply to our problem. Other works on PCA in vertical FL, which utilizes MPC alone, such as Cheung & Yu (2020); Fan et al. (2021), do not provide rigorous DP guarantees. In RMGM-OLS (Wu et al., 2018), the clients randomly project their data partitions prior to sharing, based on insights from linear regression, a supervised learning problem. Their method does not apply to ours since PCA is an unsupervised learning problem.

Most recent works on vertical FL either trust one of the clients or a third party to inject random noises into the sensitive outcome computed on the underlying data, aiming to provide DP guarantees (e.g., see Xu et al. (2021); Ranbaduge & Ding (2022)). This approach, however, violates privacy as the party could be adversarial and infer the underlying data from the sensitive outcome. The recent work by Li et al. (2023) focuses on $k$-means clustering and is orthogonal to our work. We also note that Dwork & Nissim (2004) first study private data analysis for vertically partitioned databases. However, due to the differences in problem definitions, their results are not comparable with ours.

**Our contribution.** Unlike most previous works in vertical FL, which assume the clients are trustworthy, we consider the scenario where clients are also adversarial, a problem that is much more difficult. Given such a threat model, closing the non-negligible performance gap (in terms of privacy-utility trade-offs) between the vertical FL algorithm and the corresponding centralized version in existing works (e.g., see Xu et al. (2021); Li et al. (2023)) is even more challenging. On the one hand, small local DP noises injected on the client side of FL could easily accumulate into overwhelmingly large noise, causing a steep hit in the utility (Kairouz et al., 2016). On the other hand, we cannot trust the server or a third party to perform the noise injection.

In horizontal FL, it has been shown that, with SecAgg (Bonawitz et al., 2017), one can achieve comparable performance as in the centralized setting. This approach, however, does not apply to our setting, as we have explained above. *Can we achieve comparable performance in vertical FL as in the centralized setting?* This question has remained unanswered in the literature.

We give a positive answer in this paper, by proposing SPCA, a solution for PCA over vertically partitioned data. SPCA is the first DP algorithm in vertical FL that simultaneously protects against the server and the clients while introducing noise of scale comparable with the strong centralized baseline to the result, without assuming any trusted third party. Using random matrix theory, we show that SPCA is able to achieve the optimal error rate as the centralized baseline, which is confirmed with experiments on real-world datasets.

## 2 PRELIMINARIES

### 2.1 PRINCIPAL COMPONENT ANALYSIS (PCA)

Following previous work in centralized setting (Dwork et al., 2014), we consider a dataset (matrix) $D \in \mathbb{R}^{m \times n}$ that is centered at the origin. Each row $D[i\,,:] \in \mathbb{R}^{1 \times n}$ represents the $i$-th record vector ($i \in [m]$) consisting of $n$ real-valued attributes and $\|D[i\,,:]\|_2 \leq 1$. Each column $D[:,j] \in \mathbb{R}^{m \times 1}$ represents the vector of the $j$-th attribute ($j \in [n]$) from all records. Principal component analysis (PCA) (Pearson, 1901; Jolliffe, 2014) aims at reducing the number of attributes of the dataset $D$ while capturing its covariance by performing a linear transformation. This idea turns out to be finding a rank-$k$ subspace $A \in \mathbb{R}^{n \times k}$ with the goal of maximizing the variance in $DA$, which is equivalent to finding the $k$-dimensional principal singular subspace of the covariance matrix $C = D^T D$, where

$$C[i\,,j] = \sum_{l=1}^{m} \left( D[l\,,i] \cdot D[l\,,j] \right), \tag{1}$$

for $i, j \in [n]$. Given the optimal rank-$k$ subspace $V_k$, the error of a rank-$k$ subspace $A$ on $D$ is

$$err = \|DV_k\|_F^2 - \|DA\|_F^2, \tag{2}$$

where $\|\cdot\|_F$ is the Frobenius norm and $err$ reaches 0 when the columns of $A$ match those of $V_k$.

## 2.2 Differential Privacy (DP)

Differential Privacy (DP) (Dwork et al., 2006) quantifies the indistinguishability of the output distributions for a mechanism on "neighboring" datasets, where $D$ and $D'$ are called neighboring datasets if they differ by one record (written as $D \sim D'$). A widely used DP definition is $(\epsilon, \delta)$-DP.

**Definition 1** ($(\epsilon, \delta)$-Differential Privacy (Dwork et al., 2006)). *A randomized mechanism $\mathcal{M} : \mathcal{D} \to \mathcal{Y}$ satisfies $(\epsilon, \delta)$-differential privacy (DP) if for any $\mathcal{O} \subseteq \mathcal{Y}$ it holds that*

$$\sup_{D \sim D'} \Pr[\mathcal{M}(D) \in \mathcal{O}] \leq \exp(\epsilon) \cdot \Pr[\mathcal{M}(D') \in \mathcal{O}] + \delta. \tag{3}$$

Here smaller $\epsilon$ or $\delta$ imply higher difficulty in distinguishing the distributions, and hence stronger privacy guarantees. Given a mechanism (function) $\mathcal{M} : \mathcal{D} \to \mathbb{R}^n$ of interest, a canonical approach to make it differentially private is to introduce random noise (perturbation) to its outcome. The scale of the noise is calibrated to the sensitivity of the mechanism (Dwork et al., 2006), denoted as $S(\mathcal{M})$.

$$S(\mathcal{M}) = \max_{D \sim D'} \|\mathcal{M}(D) - \mathcal{M}(D')\|, \tag{4}$$

where $\|\cdot\|$ represent a norm measure of interest (e.g. $\mathcal{L}_2$-norm). To design an effective DP algorithm, one should prefer functions with lower sensitivities regarding the input.

Rényi DP (RDP) (Mironov, 2017), an alternative definition to $(\epsilon, \delta)$-DP, is also widely used.

**Definition 2** (Rényi Differential Privacy Mironov (2017)). *A randomized mechanism $\mathcal{M}$ satisfies $(\alpha, \tau)$-Rényi differential privacy (RDP) for some $\alpha \in (0, 1) \cup (1, \infty)$ if*

$$\sup_{D \sim D'} D_\alpha(\mathcal{M}(D) \,\|\, \mathcal{M}(D')) \leq \tau,$$

*where $D_\alpha(P \,\|\, Q)$ denotes the Rényi divergence of $P$ from $Q$ that are defined over the same domain.*

We note that any mechanism that satisfies RDP also satisfies $(\epsilon, \delta)$-DP according to the conversion rules (Mironov, 2017; Canonne et al., 2020), and both DP frameworks are preserved under post-processing (Dwork et al., 2006). A commonly used perturbation mechanism in federated learning is the additive symmetric Skellam noise (denoted as $\mathrm{Sk}(\mu)$), obtained as the difference between two independent Poisson noises of parameter $\mu$, which, when applied on integer-valued functions, preserves RDP. (We will see why we focus on integer-valued functions later.)

**Lemma 1** (Skellam noise preserves RDP (Agarwal et al., 2021)). *We denote $\mathrm{Sk}^n(\mu)$ as the ensemble of $n$ independent random variables sampled from $\mathrm{Sk}(\mu)$. Then, for any integer $\alpha > 1$, and any $n$-dimensional integer-valued function $F$ with bounded $\mathcal{L}_1$ and $\mathcal{L}_2$ sensitivities, it holds that*

$$\sup_{D \sim D'} D_\alpha\left(F(D) + \mathrm{Sk}^n(\mu) \,\|\, F(D') + \mathrm{Sk}^n(\mu)\right) \leq \tau, \tag{5}$$

*where $\Delta_2 = \sup_{D \sim D'} \|F(D) - F(D')\|_2$, and $\Delta_1 = \sup_{D \sim D'} \|F(D) - F(D')\|_1$, and*

$$\tau = \frac{\alpha \Delta_2^2}{4\mu} + \min\left(\frac{(2\alpha - 1)\Delta_2^2 + 6\Delta_1}{16\mu^2}, \frac{3\Delta_1}{4\mu}\right).$$

A notable property of Skellam is that the summation of independent Skellam noises still follows the Skellam distribution. We will see how this property comes into play in our solution in Section 4.

## 2.3 Secure Multiparty Computation (MPC)

Broadly speaking, secure multiparty computation (MPC) (Yao, 1986; Chaum et al., 1987) coordinates multiple parties to jointly compute a function without revealing their inputs. Due to this security property (namely, the computation process does not leak information about the private inputs), MPC has been widely applied in federated learning (e.g., see Bonawitz et al. (2017); Mohassel & Zhang (2017)). In the case of PCA, the function of interest is the covariance matrix as in Eq. 1, and the private inputs are the local partitions of the matrix $D$.

To compute such an outcome with MPC, we take the classic BGW protocol (Ben-Or et al., 1988) that is based on Shamir's secret sharing (SSS) (Shamir, 1979) as an example. SSS enables a secret holder to distribute a secret among a group of parties in a way such that: 1) no single party is able to learn any non-trivial information about the secret; 2) when a sufficiently large number of parties combine their information, the secret is *reconstructed*. BGW, which is built upon SSS, implements a three-phase execution for computing a sensitive function.

1. First, each party distributes her private input as secret shares to other clients using Shamir's algorithm (Shamir, 1979). In our setting, each party distributes her private data partition to other clients as secret shares. Note that no party can infer any other party's private data partition, which is enforced by the security property of Shamir's algorithm.

2. Next, each party simulates the computation of the function (e.g., the covariance matrix as in Eq. 1) using a digital circuit while keeping the value of each computed gate (of the circuit) as secret shared by all parties. Similar to the first step, the value of each computed gate can not be inferred by any party.

3. Finally, all of the parties reconstruct the true outcome of the function, using their secret shares. This whole process is repeated independently for every entry in the upper diagonal of the covariance matrix (in total there are $n(n+1)/2$ entries).

As we have mentioned, the output of MPC, however, still leaks information about the private inputs. For example, seeing that $C[i, j] = 0$ means that there is no linear relationship between the empirical distribution of the $i$-th attribute and the $j$-th attribute in $D$, which is considered sensitive information. This motivates us to consider differential privacy, which protects the underlying data from being inferred from the outcome.

## 3 PROBLEM DEFINITION AND CHALLENGES

### 3.1 PRINCIPAL COMPONENT ANALYSIS IN VERTICAL FEDERATED LEARNING

In our setting, each of the clients possesses a collection of attributes of the dataset $D \in \mathbb{R}^{m \times n}$ (i.e., a subset of columns in $D$). More specifically, we assume there are $N$ clients, where client $q$ possesses a portion $D_q$ of $D$ ($q \in [N]$). Here $D_q$ consists of a collection of attributes $\mathcal{A}_q$ such that $D_q = \{D[:, j] : j \in \mathcal{A}_q\}$. Besides, we assume that the collections of columns (resp. attributes) $\{D_q : q = 1, \ldots, N\}$ (resp. $\{\mathcal{A}_q : q = 1, \ldots, N\}$) are mutually exclusive. Namely, $\sum_{q=1}^{N} |\mathcal{A}_q| = n$.

Given an integer $k$ (usually $k \ll n$), an untrusted server aims to (approximately) learn a rank-$k$ subspace $\tilde{V}_k \in \mathbb{R}^{n \times k}$ that preserves most of the variance in the original dataset $D$, namely, maximizing $\|D\tilde{V}_k\|_F^2$. The goal of this work is to design a mechanism $\mathcal{M}$ for finding such $\tilde{V}_k$ of high utility while preserving differential privacy (formalized next).

### 3.2 PRIVACY REQUIREMENTS

Recall that there are two types of parties involved in our setting, a client $q$ and the untrusted server. For a randomized mechanism $\mathcal{M}$, we use $\mathcal{M}_{\text{server}}$ and $\mathcal{M}_{\text{client}_q}$ to represent the observations by the server and a client $q$ out of $\mathcal{M}$, respectively (we will explain the difference later). Accordingly, we need to consider two levels of privacy, server-observed DP, and client-observed DP. In what follows, we explain the privacy requirements in more detail under the standard $(\epsilon, \delta)$-DP.

To protect individual records in $D$ from being inferred by the adversarial server, we consider server-observed DP, as is done in previous works of horizontal federated learning.

**Definition 3** (server-observed DP). *A randomized mechanism $\mathcal{M}$ satisfies $(\epsilon, \delta)$-server-observed DP if for any set of output $\mathcal{O} \subseteq Range(\mathcal{M}_{server})$, it holds that*

$$\sup_{D \sim D'} \Pr[\mathcal{M}_{server}(D) \in \mathcal{O}] \leq \exp(\epsilon) \cdot \Pr[\mathcal{M}_{server}(D') \in \mathcal{O}] + \delta.$$

*Here "$\sim$" represents the neighboring relation of two datasets (resp. matrices) that one can be obtained from the other by adding/removing one record (resp. row).*

As a participant, an adversarial client may also try to infer other clients' local portions from the outputs of $\mathcal{M}$. To prevent this from happening, we consider client-observed DP.

**Definition 4** (client-observed DP). *A randomized mechanism $\mathcal{M}$ satisfies $(\epsilon, \delta)$-client-observed DP if for every client $q$ and any set of output $\mathcal{O} \subseteq Range(\mathcal{M}_{client_q})$, it holds that*

$$\sup_{D \sim D'} \Pr[\mathcal{M}_{client_q}(D) \in \mathcal{O}] \leq \exp(\epsilon) \cdot \Pr[\mathcal{M}_{client_q}(D') \in \mathcal{O}] + \delta.$$

*Here "$\sim$" represents the neighboring relation of two datasets (resp. matrices) that one can be obtained from the other by replacing one record (resp. row).*

We explain the differences in Definitions 3 and 4 next. First, in federated learning, the knowledge of a client differs from the server's. To see this, note that every client $q$ has full access to the corresponding private portion $D_q$ (including the information for identifying an individual record). On the other hand, the server does not have access to the dataset $D$ (including the number of records in $D$). Hence, for client-observed DP, we have adopted the bounded-DP definition (Dwork et al., 2006), where the size of $D$ (namely, $m$) is not considered private information. In the meantime, for server-observed DP, we have adopted the more widely used unbounded DP definition (Dwork, 2006), which also protects the information of $m$ from the server, as is done in previous works in FL (Kairouz et al., 2021).

Second, the observation of a client (namely, $\mathcal{M}_{\text{client}_q}(D)$) may also differ from the server's (namely, $\mathcal{M}_{\text{server}}(D)$). As a result, one cannot obtain the privacy parameters of client-observed DP by implying multiplying those of server-observed DP by two, as this transformation relies on the triangle inequality regarding the divergence of the random variables over the *exact same domain*. In general, there is no conversion rule between server-observed DP and client-observed DP. Hence, it is necessary to have both Definitions 3 and 4, if we consider both server and clients adversarial. To give a more concrete example, we show that a mechanism that preserves server-observed DP can be blatantly non-private for a client as well as its reverse statement.

**Proposition 1.** *A mechanism $\mathcal{M}$ that satisfies* server-observed *DP can be non-private for a client.*

*Proof.* We prove by a construction of such an $\mathcal{M}$. Consider the simple randomized algorithm that estimates the sum of a dataset $\sum_{\mathbf{x} \in D} \mathbf{x}$. The algorithm starts with all clients sending their private partition directly (i.e. without perturbation) to some client $k$, without loss of generality, assuming $k = 1$, who then computes the sum $\sum_{\mathbf{x} \in D} \mathbf{x}$; obtains $\tilde{x}$ by perturbing the sum with Gaussian noise; and sends the outcome $\tilde{x}$ to the server. This mechanism satisfies server-observed DP, since the observation of the server is perturbed by additive Gaussian noise (Dwork & Roth, 2014; Balle & Wang, 2018). However, for client 1, who observed the data partitions collected from all other clients, this mechanism is non-private. $\square$

**Proposition 2.** *A mechanism $\mathcal{M}$ that satisfies* client-observed *DP can be non-private for the server.*

*Proof.* The proof is similar to the above one. We consider the mechanism where every client first sends her local data to the server, who then computes the sum the sum $\sum_{\mathbf{x} \in D} \mathbf{x}$; obtains $\tilde{x}$ by perturbing the sum with Gaussian noise; and broadcasts the outcome $\tilde{x}$ to the clients. The client-observed DP comes from the Gaussian noise whereas for the server, the mechanism is non-private. $\square$

In this paper, we focus on the semi-honest threat model, where an adversary (either a server or a client) strictly follows the execution of $\mathcal{M}$ while trying to infer the private data in other clients' partitions. The settings where the clients can be malicious are considered future works. We also assume there is an error-free and secure channel between each pair of clients. Both aforementioned assumptions have been adopted in previous works (Kairouz et al., 2021; Agarwal et al., 2021).

### 3.3 BASELINE SOLUTIONS AND CHALLENGES

**Baseline in the centralized setting.** We first recall the optimal solution in the *centralized* setting (Dwork et al., 2014), where a trusted data curator, who possesses the entire matrix $D$, injects independent Gaussian noises to the upper diagonal of the covariance matrix $D^T D$. For each $i, j \in [n]$ and $i \le j$, the data curator computes

$$\hat{C}[i, j] = \sum_{l=1}^{m} (D[l, i] \cdot D[l, j]) + z, \tag{6}$$

where $z$ is sampled from $\mathcal{N}(0, \sigma^2)$ and $\sigma$ is determined by the privacy constraints. Matrix $\hat{C}$ is then released to the analyst/adversary, who could compute the $k$-dimensional singular subspace using standard non-DP algorithms such as Klema & Laub (1980). The privacy guarantee follows from the DP guarantee of additive Gaussian noises and the post-processing property of DP.

**Challenges in vertical FL.** To perform the noise injection as in Eq. 6 while respecting both levels of DP, however, is rather challenging in our setting. This is because the computation of

$\sum_{l=1}^{m} (D[l,i] \cdot D[l,j])$ already contains sensitive information of different clients' submatrices. Allowing a single party (a client or a server) to add random noise $z$ to such an outcome violates privacy, as this party knows the value of $z$ and could infer the clean outcome from the perturbed version (this approach was adopted in, e.g., Xu et al. (2021)). Wu et al. (2020) let the clients *jointly* sample random noise $z$ such that no client knows the exact value of the noise. However, they did not provide any quantifiable DP guarantee on the client side.

**Baseline in vertical FL.** The baseline solution that respects both levels of DP is as follows. First, each client independently perturbs her local data portion (namely, a submatrix) with some random Gaussian noise. After perturbation, all the clients reveal their perturbed submatrices to one of the clients, who then reconstructs the whole perturbed matrix; computes the covariance matrix from the perturbed matrix; computes the $k$-dimensional singular subspace using standard non-DP algorithm; and sends the result to the server. The overall amount of noise injected into every entry of the covariance matrix is as large as $O(m)$, which ultimately leads to unsatisfactory utility.

## 4 OUR SOLUTION AND ANALYSIS

**Idea.** Our idea is to let each client independently inject additive Skellam noise into the computation of each upper-diagonal entry in the covariance matrix $C = D^T D$. In addition, this whole process, including *noise injection* and *covariance computation*, is done using the general BGW protocol as a black box. Combined with the security property of MPC, the aggregation of additive Skellam ensures server-observed DP and client-observed DP at the same time. We ignore the fact that $D$ is partitioned among clients for the moment. To protect data privacy while maintaining high utility, the random perturbation to achieve DP should be applied *directly* on $C = D^T D$ as is done in Eq. 6. To achieve server-observed DP, the random perturbation $z$ should come from the client side, instead of the server side. To achieve client-observed DP, no single client should be able to learn the real value of $z$. This also rules out the option of allowing a single client to inject the DP noise), suggesting the following approach.

**Independent noise contributions.** To perturb an entry in $C$, each client $q$ independently contributes her random perturbation, denoted as $z_q$, to the outcome. Independent $z_q$ sums up to a larger random perturbation, written as $\sum_{q \in [N]} z_q$, such that no client alone learns its true value. In the meantime, the overall privacy guarantee provided by $z = \sum_{q \in [N]} z_q$ is (roughly) $N$ times stronger than what each $z_q$ provided. As we will see in the analysis later, there is a slight difference between the server-observed privacy and the client-observed privacy.

**The discrete Skellam noise.** Secure multiparty computation allows each client to perform the above noise injection in collaboration with other clients without revealing her inputs to others (including noise $z_q$ as well as the local data). Since MPC algorithms operate over discrete integers (more precisely, finite field elements), a natural choice for the noise distribution of $z_q$ is the discrete Skellam noise, instead of the classic Gaussian or Laplace noise. In addition, the sum of independent Skellam noises $\sum_{q \in [N]} z_q$ still follows a Skellam distribution and can be analyzed with Lemma 1.

**The algorithm.** We outline SPCA as in Algorithm 1, described in more detail as follows. For now, we focus on a pair of indices $i, j \in [n]$ such that $i \leq j$ and $D[,i]$ and $D[,j]$ are possessed by two different clients, say clients $p$ and $k$. In what follows, we describe how to approximate $C[i,j]$ in a privacy-preserving manner. First, client $p$ and $k$ independently discretize every element in $D[,i]$ and $D[,j]$ using Algorithm 2 (upscaling followed by rounding). This step converts the data from the continuous domain into the discrete domain, setting up the stage for MPC algorithms. Next, each client $q \in [N]$ generates her DP noise for perturbing this covariance entry. We denote the noise as $z_q[i,j]$, which is sampled from $Sk(\mu/N)$. After that, all the clients collaboratively compute

$$\tilde{C}[i,j] = \sum_{l=1}^{m} \overline{D}[l,i] \cdot \overline{D}[l,j] + \sum_{q \in [N]} z_q, \qquad (7)$$

using the BGW protocol. For clients $p$ (resp. $k$), the inputs to BGW are the $i$-th (resp. $j$-th) column of the discretized matrix $\overline{D}$ and the Skellam noise $z_p$ (resp. $z_k$). For any other client $q \in [N]$ and $q \neq p, k$, the input to BGW is the Skellam noise $z_q$. Within in BGW protocol, each party (namely, a client) distributes her input(s) as secret shares to all the clients such that no client alone is able to

---

**Algorithm 1:** SPCA

---

**Input:** Private matrix $D \in \mathbb{R}^{m \times n}$; number of clients $N$, discretization parameter $\gamma$, noise parameter $\mu$.
1 **for** *every client* $q \in [N]$ **do**
2     Client $q$ independently discretizes every entry in her local partition (submatrix) using Algorithm 2.

3 We denote the discretized data matrix as $\bar{D}$.
4 **for** *every row* $i \in [n]$ **do**
5     **for** *every column* $j \in [n]$ *and* $i \leq j$ **do**
6        **if** *columns* $\overline{D}[\,,i]$ *and* $\overline{D}[\,,j]$ *are located in different clients* **then**
7           **for** *every client* $q \in [N]$ **do**
8              Client $q$ samples $z_q$ independently from $Sk(\mu/N)$.
9           *All clients* compute $\tilde{C}[i,j] = \sum_{l=1}^{m} \overline{D}[l\,,i] \cdot \overline{D}[l\,,j] + \sum_{q \in [N]} z_q$ using BGW protocol.
10        **else**
11           *The client* with columns $\overline{D}[\,,i]$ and $\overline{D}[\,,j]$ computes $\tilde{C}[i,j] = \sum_{l=1}^{m} \overline{D}[l\,,i] \cdot \overline{D}[l\,,j] + z$,
             where $z$ is sampled from $Sk(\mu)$.

12 The server obtains $\tilde{C}$ from the clients and reconstructs a symmetric matrix using its upper diagonal entries.
13 The server computes $\tilde{V}_k$, the $k$-dimensional principal singular subspace (i.e., top $k$ eigenvectors) of $\frac{1}{\gamma^2} \cdot \tilde{C}$.
**Output:** $\tilde{V}_k$.

---

**Algorithm 2:** Discretization Procedure

---

**Input:** Value to discretize $x \in \mathbb{R}$, discretization parameter $\gamma$
1 $u \leftarrow \gamma \cdot x$.
2 Flip a coin with heads probability $u(j) - \lfloor u(j) \rfloor$.
3 **if** *Heads* **then**
4     $v(j) \leftarrow \lfloor u(j) \rfloor + 1$.
5 **else**
6     $v(j) \leftarrow \lfloor u(j) \rfloor$.
**Output:** Integer $u \in \mathbb{Z}$.

---

learn the true values of the Skellam noise and the discretized data[1]. After the execution of BGW, the clients share the outcome matrix with the server. The server then reconstructs the covariance matrix from its upper diagonal elements; downscales the matrix; and finally computes the $k$-dimensional singular subspace using standard non-DP algorithms.

When $D[\,,i]$ and $D[\,,j]$ are possessed by a single client, she simply injects noise sampled from $Sk(\mu)$ into $\sum_{l=1}^{m} \overline{D}[l\,,i] \cdot \overline{D}[l\,,j]$ computed on her side, and then sends the perturbed outcome to the server.

**On discretization.** We have used an explicit discretization procedure (Algorithm 2) for pre-processing the continuous data instead of using a black box (such as the $64$-bit representation of double-precision floating numbers). The reason is that we want to accurately account for the sensitivity that is crucial for DP analysis, since careless computation may cause privacy violation (e.g., see Casacuberta et al. (2022)). In terms of the discretization parameter $\gamma$, we suggest setting $\gamma = O(n)$ to make sure that the error due to discretization does not overwhelm the error due to DP perturbation. Intuitively, since the record is normalized to 1, we need a larger $\gamma$ as the dimension ($n$) increases to avoid loss of information during discretization. We would also like to emphasize that we cannot use the continuous noise (e.g., Gaussian (Ranbaduge & Ding, 2022)) in Line 11 of Algorithm 1, since the summation of discretized Gaussians (communicated in FL) does not provide a tractable privacy guarantee.

**Overheads.** A discretization parameter of $\gamma$ translates into $\log \gamma$ bits per dimension for communication. In our experiments, the number of bits never exceeds $14$, which is smaller than the commonly used 32 bit representation in modern computers and FL applications. Finally, both procedures for discretizing the data and the generation of Skellam can be done in parallel efficiently and the additional communication cost for injecting noises is $O(Nn^2)$, which is smaller than the $O(mn^2)$

---

[1]We can replace the BGW protocol with any other MPC protocol (e.g., SPDZ (Keller et al., 2018)), without affecting the privacy guarantees.

communication cost for computing the covariance matrix in FL considering that $N < n$. Namely, the overhead due to DP is not significant, and SPCA can be deployed into the real world efficiently.

**Federated Analytics.** To free the clients from repeatedly participating in MPC, which requires synchronicity, we can adapt SPCA to the anonymous aggregation framework (Talwar et al., 2023) of federated analytics (FA) (Google Research, 2020) with the following modifications. Instead of performing the MPC protocol on the client side, the clients can first distribute their data and noises as secret shares with non-colluding servers, who then collaboratively perform the computation as in Eq. 7 using MPC. Note that the original FA only considers linear functions, and our solution can be seen as a non-linear extension. We also extend SPCA to logistic regression in the Appendix.

### 4.1 ANALYSIS

We first show that our solution SPCA satisfies Rényi-DP, regarding both the server and a client.

**Lemma 2.** *Given discretization parameter $\gamma$ and noise parameter $\mu$, Algorithm 1 satisfies $(\alpha, \tau_{server})$-*server-observed *RDP with*

$$\tau_{server} = \frac{\alpha \Delta_2^2}{4\mu} + \min\left(\frac{(2\alpha - 1)\Delta_2^2 + 6\Delta_1}{16\mu^2}, \frac{3\Delta_1}{4\mu}\right), \tag{8}$$

*and satisfies $(\alpha, \tau_{client})$-*client-observed *RDP with*

$$\tau_{client} = \frac{\alpha N^2 \Delta_2^2}{(N-1)^2 \mu} + \min\left(\frac{Nn^2}{(N-1)^2} \cdot \frac{(2\alpha - 1)\Delta_2^2 + 3\Delta_1}{4\mu^2}, \frac{3n\Delta_1}{2(N-1)\mu}\right), \tag{9}$$

*where $\Delta_2 = \gamma^2 + n$ and $\Delta_1 = \min((\gamma^2 + n)^2, \sqrt{n}(\gamma^2 + n))$.*

The corresponding $(\epsilon, \delta)$-DP guarantees can be obtained using the conversion rules. To understand the privacy guarantee, note that: 1) for the server, the overall DP noise in every entry of $\tilde{C}$ is effectively sampled from $\mathrm{Sk}(\mu)$, and 2) for a client, the noise is effectively sampled from $\mathrm{Sk}(\frac{N-1}{N}\mu)$, since every client $q$ already observes her own share of DP noise and the security property of MPC prevents the client from learning other clients' inputs, including the local Skellam noises. Finally, the difference in defining neighboring datasets contributes to the additional factor(s) of 4 in Eq. 8.

Lemma 3 formalizes the error of SPCA under $(\epsilon, \delta)$-DP (the analysis under RDP is in the Appendix).

**Lemma 3.** *Let $V_k$ be the rank-$k$ subspace of the original matrix $D$ and let $\tilde{V}_k$ be the principal rank-$k$ subspace of matrix $\tilde{C}$ obtained from Algorithm 1 with discretization parameter $\gamma = O(n)$. Then Algorithm 1 satisfies $(\epsilon, \delta)$-*server-observed *DP and with high probability, we have that*

$$\|D\tilde{V}_k\|_F^2 \geq \|DV_k\|_F^2 - O(k\sqrt{n}\sqrt{\mu_{\epsilon,\delta}}), \tag{10}$$

*where $\sqrt{\mu_{\epsilon,\delta}} = c\sqrt{\log(1/\delta)}/\epsilon$ for some constant $c$.*

The error rate of Eq. 10 is independent of $N$, matching the optimal error rate of Dwork et al. (2014) in the centralized setting (we defer the detailed discussion on $N$ to Appendix A.3). On the other hand, the error of the distributed baseline is $O(mkn \log(1/\delta)/\epsilon^2)$ (we will see the performance gap next).

## 5 EXPERIMENTS

**Setup.** We compare our solution with the centralized-DP solution Dwork et al. (2014) and the distributed baseline described in Section 3.3 (abbreviated as "Centralized" and "Baseline" in the figures, respectively). In particular, the performance of the strong centralized algorithm (Dwork et al., 2014) is regarded as the goal (the upper limit) for our SPCA to achieve. We conduct experiments on the KDDCUP dataset ($m = 195666$ and $n = 117$) (Tavallaee et al., 2009), the ACSIncome dataset of California ($m = 145585$ and $n = 816$) (Ding et al., 2021), the CiteSeer dataset ($m = 2110$ and $n = 3703$) (Giles et al., 1998), and the Gene dataset ($m = 801$ and $n = 20531$) (Fiorini, 2016). We report the utility of $\tilde{V}_k$ computed on the input dataset $D$ for different values of $k$, defined as $\|D\tilde{V}_k\|_F^2$. For datasets of higher dimensions, we choose larger $k$'s. In terms of privacy, we focus on the server-observed DP under the standard $(\epsilon, \delta)$-DP framework, since it is also commonly considered in the centralized and horizontal FL settings. We fix $\delta$ to $10^{-5}$ and vary $\epsilon$. For our solution

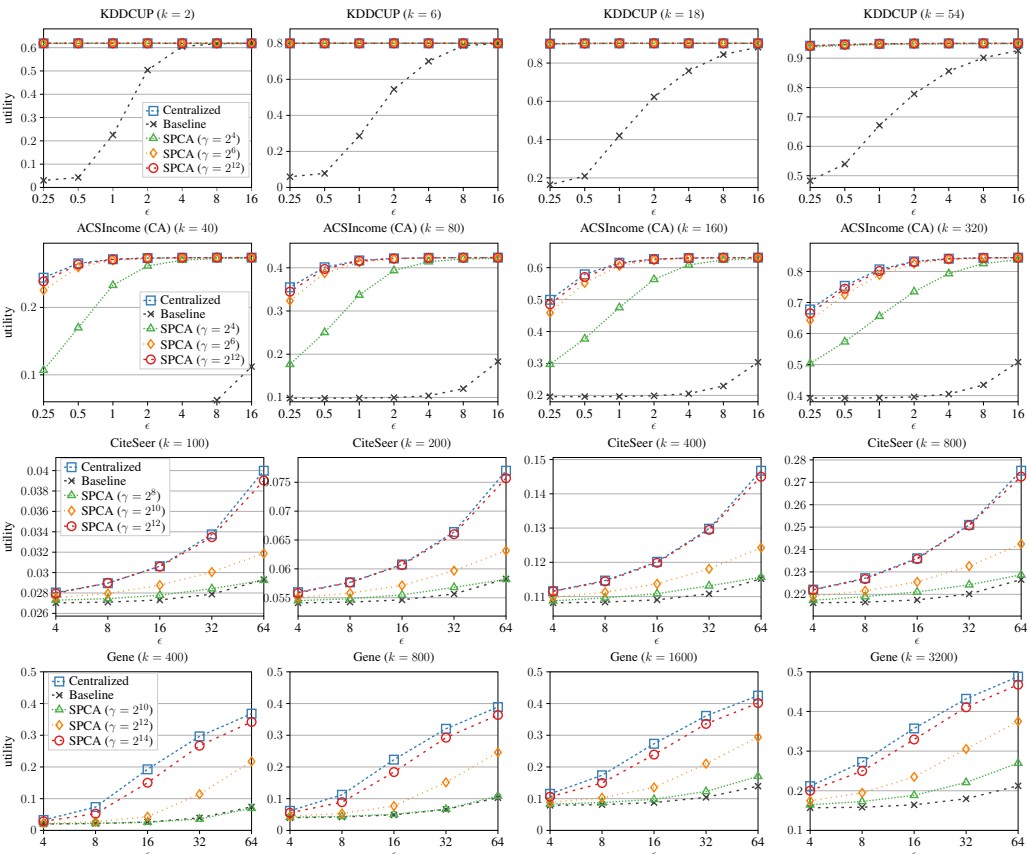

Figure 1: Performance on multiple datasets under different values of $\epsilon$ with $\delta$ fixed to $10^{-5}$.

SPCA, we also vary the discretization parameter $\gamma$. For the high-dimensional Gene and CiteSeer datasets, we choose relatively larger $\gamma$ to avoid loss of information during discretization and also larger $\epsilon$ to maintain the signal-noise ratio . We fix $\delta$ to $10^{-5}$. As the number of clients $N$ does not affect the server-observed DP guarantee, we fix $N$ to the number of columns in $D$ (i.e., $n$) without loss of generality.

**Results.** We report the average utility over 20 independent runs in Figure 1, from which it is clear that our solution always outperforms the distributed baseline and is close to the strong centralized-DP baseline under various parameter settings (in particular, when $\gamma$ is large enough), confirming the efficacy of SPCA. Note that the standard deviation for the utility SPCA is constantly less than 0.001 in our experiments. For KDDCUP, SPCA performs almost the same as the strong centralized-DP baseline for a wide range of $\gamma$. For ACSIncome, the difference between SPCA and the centralized baseline is negligible when $\gamma \geq 2^6$, and we also note that even when $\gamma = 2^4$, SPCA still significantly outperforms the distributed baseline. Similar conclusions also apply to other two high-dimensional datasets. For Gene and CiteSeer. the performance gap between SPCA and the centralized solution is negligible when $\gamma$ is large enough (when $\gamma$ reaches $2^{14}$ and $2^{12}$, respectively). The improvement over the baseline is also significant under large enough $\gamma$'s, confirming the efficacy of SPCA.

## 6 CONCLUSION

In this work, we study principal component analysis in vertical federated learning with differential privacy constraints. We present the first solution in the literature that simultaneously enforces two levels of DP without assuming any trusted third party. We show that our solution achieves privacy-utility trade-offs matching the centralized setting through theoretical analysis and empirical evaluations. Regarding future work, we plan to extend our idea to applications in vertical FL beyond PCA and also study the setting where clients are malicious.

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

# A APPENDIX

## A.1 ADDITIONAL BACKGROUND ON DIFFERENTIAL PRIVACY

Injecting Gaussian noise to a function $F$ satisfies both Rényi-DP and $(\epsilon, \delta)$-DP, formalized as follows.

**Lemma 4** (Gaussian Mechanism Preserves RDP (Mironov, 2017)). *Injecting Gaussian noise sampled from $\mathcal{N}\left(\mathbf{0}, \sigma^2 \cdot \mathbf{I}\right)$ to function $F$ of sensitivity $S(F)$ satisfies $(\alpha, \frac{\alpha S^2(F)}{2\sigma^2})$-RDP.*

**Lemma 5** (Analytic Gaussian Mechanism (Balle & Wang, 2018)). *Injecting Gaussian noise $\mathcal{N}\left(\mathbf{0}, \sigma^2 \cdot \mathbf{I}\right)$ into the output of $F$ satisfies $(\epsilon, \delta)$-differential privacy, if*

$$\frac{S(F)}{\sigma} \leq \sqrt{2}\left(\sqrt{\chi^2 + \epsilon} - \chi\right),$$

*where $\mathbf{0}$ and $\mathbf{I}$ are a zero vector and a $d \times d$ identity matrix, respectively, and $\chi$ is the solution to*

$$\operatorname{erfc}(\chi) - \exp(\epsilon) \cdot \operatorname{erfc}\left(\sqrt{\chi^2 + \epsilon}\right) = 2\delta,$$

*and $\operatorname{erfc}()$ denotes the complementary error function, i.e., $\operatorname{erfc}(x) = 1 - \frac{2}{\sqrt{\pi}} \int_0^x e^{-t^2}\, \mathrm{d}t$.*

If a mechanism is Rényi-DP, then the same mechanism that runs on a random subset of the input dataset also satisfies Rényi-DP.

**Lemma 6** (Subsampling for RDP (Zhu & Wang, 2019; Mironov et al., 2019)). *Let $\mathcal{M}$ be a mechanism that satisfies $(l, \tau_l)$-RDP for $l = 2, \ldots, \alpha$ ($\alpha \in \mathbb{Z}, \alpha \leq 2$), and $S_q$ be a procedure that uniformly samples each record of the input data with probability $q$. Then $\mathcal{M} \circ S_q$ satisfies $(\alpha, \tau)$-RDP with*

$$\tau = \frac{1}{\alpha - 1} \cdot \log\left((1-q)^{\alpha-1}(\alpha q - q + 1) + \sum_{l=2}^{\alpha}\binom{\alpha}{l}(1-q)^{\alpha-l}q^l e^{(l-1)\tau_l}\right).$$

If a mechanism is Rényi-DP, then repeatedly running the same mechanism also satisfies Rényi-DP.

**Lemma 7** (Composition Lemma (Mironov, 2017)). *If mechanisms $\mathcal{M}_1, \ldots, \mathcal{M}_T$ satisfies $(\alpha, \tau_1), \ldots, (\alpha, \tau_T)$-RDP, respectively, then $\mathcal{M}_1 \circ \ldots \circ \mathcal{M}_T$ satisfies $(\alpha, \sum_{t=1}^{T}\tau_i)$-RDP.*

Post-processing (either randomized or deterministic), performed after a differentially private mechanism, does not affect the privacy guarantee, according to the following lemma.

**Lemma 8** (Post-processing for DP (Dwork & Roth, 2014; van Erven & Harremoës, 2014; Mironov, 2017)). *Let $\mathcal{M}$ be an $(\epsilon, \delta)$-differentially private (resp. $(\alpha, \tau)$-RDP) mechanism, and $G$ be a function whose input is the output of $\mathcal{M}$. Then, $G(\mathcal{M})$ also satisfies $(\epsilon, \delta)$-DP (resp. $(\alpha, \tau)$-RDP).*

Finally, we present the conversion rule from Rényi-DP to $(\epsilon, \delta)$-DP.

**Lemma 9** (Converting $(\alpha, \tau)$-RDP to $(\epsilon, \delta)$-DP (Canonne et al., 2020)). *Given a mechanism $\mathcal{M}$ satisfies $(\alpha, \tau)$-RDP for any $\alpha \in (1, \infty)$, it satisfies $(\epsilon, \delta)$-DP for $\delta > 0$ and*

$$\epsilon = \tau + \frac{\log(1/\delta) + (\alpha - 1)\log(1 - 1/\alpha) - \log(\alpha)}{\alpha - 1}.$$

## A.2 BASELINE SOLUTION OF DP PCA FOR VERTICALLY PARTITIONED DATA

The baseline solution of DP PCA for vertically partitioned data (Algorithm 3) is outlined as follows. Each client first independently injects Gaussian noise into each entry of her local partition. Next, the clients share the perturbed local partitions with one of the clients (say, client 1), who then reconstructs the perturbed full matrix using the perturbed partitions and performs a non-private PCA algorithm on the perturbed matrix.

Algorithm 3 satisfies both server-observed DP and client-observed DP.

**Lemma 10.** *For any fixed $\delta_{server} \in (0, 1)$ and $\delta_{client} \in (0, 1)$, and noise parameter $\sigma$, Algorithm 3 satisfies $(\epsilon_{server}, \delta_{server})$-server-observed DP and $(\epsilon_{client}, \delta_{client})$-client-observed DP where $\epsilon_{server}$ and $\epsilon_{client}$ are the solutions to*

$$\frac{1}{\sigma} = \sqrt{2}\left(\sqrt{\chi_{server}^2 + \epsilon_{server}} - \chi_{server}\right), \tag{11}$$

---

**Algorithm 3:** Baseline Solution for DP PCA in Vertical FL

---

**Input:** Private matrix $D \in \mathbb{R}^{m \times n}$, where the $\mathcal{L}_2$ norm of each row is bounded by 1; number of clients $N$, noise parameter $\sigma$.

1 **for** *each client $q \in [N]$* **do**
2   Client $q$ perturbs every entry in $D_q$ with independent Gaussian noise sampled from $\mathcal{N}(0, \sigma^2)$, obtaining $\tilde{D}_q$.
3   Client $q$ reveals her perturbed partition $\tilde{D}_q$ to client 1.
4 Client 1 reconstructs the matrix $\tilde{D}$.
5 Client 1 computes the optimal rank-$k$ subspace of $\tilde{D}$ on her side and sends the outcome to the server.

---

*and*

$$\frac{2}{\sigma} = \sqrt{2}\left(\sqrt{\chi_{client}^2 + \epsilon_{client}} - \chi_{client}\right), \tag{12}$$

*respectively. Here $\chi_{server}$ and $\chi_{client}$ are the solutions to*

$$\mathrm{erfc}\left(\chi_{server}\right) - \exp(\epsilon_{server}) \cdot \mathrm{erfc}\left(\sqrt{\chi_{server}^2 + \epsilon_{server}}\right) = 2\delta_{server}, \; and$$

$$\mathrm{erfc}\left(\chi_{client}\right) - \exp(\epsilon_{client}) \cdot \mathrm{erfc}\left(\sqrt{\chi_{client}^2 + \epsilon_{client}}\right) = 2\delta_{client}, \; respectively.$$

The proof follows from Lemma 5 and Lemma 8. Note that the numerator on the left-hand sides of Eq. 12 and Eq. 11 defer by a factor of 2. This is because, in client-observed DP, the sensitivity is two times that in server-observed DP since the clients already know the number of records $m$, as we have mentioned in Section 3 of the main paper.

## A.3 DETAILED ANALYSIS FOR SPCA

We first review a fact of the discretization procedure (Algorithm 2). Given discretization parameter $\gamma$ for the discretization algorithm, if each element of the vector $u \in \mathbb{R}^n$ is discretized using Algorithm 2, then we have the following inequality on the $\mathcal{L}_2$ norm of the discretized vector $\overline{u}$

$$\|\overline{u}\|_2 \leq \sqrt{\gamma^2 \cdot \|u\|_2^2 + n}. \tag{13}$$

In our setting, we assume $\|u\|_2 \leq 1$, then each row in the discretized matrix $\bar{D}$ has bounded $\mathcal{L}_2$ norm of $\sqrt{\gamma^2 + n}$. Next, we present the proof for Lemma 2 in the main paper.

*Proof of Lemma 2.* To analyze the privacy guarantees for SPCA, it suffices to analyze the privacy guarantees for computing $\tilde{C}$ regarding the discretized matrix $\overline{D}$. This is because if the computation for $\tilde{C}$ leaks no information about $\overline{D}$, then it must leak no information about $D$ since otherwise, we can infer $\overline{D}$ from $D$. In other words, $D \to \overline{D} \to \tilde{C}$ forms a Markov chain. We first focus on server-observed DP. The matrix $\tilde{C}$ observed by the server is obtained by perturbing the upper diagonal entries of $\overline{D}^T\overline{D}$ with a symmetric random matrix $E$, where each entry of $E$ follows $\mathrm{Sk}(\mu)$, since each client contributes a share of $\mathrm{Sk}(\mu/N)$ to the random perturbation. Next, we show that the server can not distinguish between $\overline{D}^T\overline{D} + E$ and $\overline{D}'^T\overline{D}' + E$, for any neighboring datasets $\overline{D}$ and $\overline{D}'$ that $\overline{D}$ can be obtained from $\overline{D}'$ by adding/removing one row. Without loss of generality, we consider $D$ has $m$ rows and $D'$ is obtained by removing the $m$-th row from $D$.

Since the $\mathcal{L}_2$ norm for each row in $\overline{D}$ and $\overline{D}'$ is bounded by $\sqrt{\gamma^2 + n}$ (the additive $n$ is due to rounding), the following holds for every $\overline{D}$ and $\overline{D}'$

$$\|\overline{D}^T\overline{D} - \overline{D}'^T\overline{D}'\|_F = \sqrt{\sum_{i,j}\left(\sum_{k=1}^{m}\overline{D}[k,i]\overline{D}[k,j] - \sum_{k=1}^{m-1}\overline{D}'[k,i]\overline{D}'[k,j]\right)^2}$$

$$= \sqrt{\sum_{i,j}\left(\overline{D}[m,i]\overline{D}[m,j]\right)^2}$$

$$= \|\overline{D}[m,:]\|_2^2$$

$$\leq \gamma^2 + n.$$

In addition, the $\mathcal{L}_1$ norm of a $d$-dimensional integer-valued vector $v$ is always less than or equal to both $\|v\|_2^2$ and $\sqrt{d}\|v\|_2$. Namely, $\|v\|_1 \leq \min(\|v\|_2^2, \sqrt{d}\|v\|_2)$ for $v \in \mathbb{Z}^d$. The proof then follows from Lemma 1 by plugging in $\Delta_2 = \gamma^2 + n$, $\Delta_1 = \min(\Delta_2^2, \sqrt{d}\Delta_2) = \min((\gamma^2+n)^2, \sqrt{n}(\gamma^2+n))$. The proof for client-observed DP is similar, expect that the $\mathcal{L}_1$ and $\mathcal{L}_2$ sensitivities are doubled and $\mu$ is replaced with $\frac{N-1}{N}\mu$. This is because, due to the security property of BGW (as well as other MPC protocols), each client $q$ only learns the outcome $\tilde{C}$ and the SKellam noises she herself generated. As a result, from a client's perspective, each entry in $\tilde{C}$ is perturbed by $Sk(\frac{N-1}{N}\mu)$. In addition, the sensitivities of the covariance matrix are doubled, since in client-observed DP, a neighboring dataset is obtained by replacing a row (changing up to $n_a$ elements/attributes) of the other dataset. $\qquad\square$

Before we present the proof of utility guarantee of SPCA (Lemma 3), we first prove the following.

**Lemma 11.** *Let $V_k$ be the rank-$k$ subspace of the original matrix $D$ and let $\tilde{V}_k$ be the principal rank-$k$ subspace of matrix $\tilde{C}$ obtained from Algorithm 1 with discretization parameter $\gamma = O(n)$ and noise parameter $\mu$. Then with high probability, we have that*

$$\|D\tilde{V}_k\|_F^2 \geq \|DV_k\|_F^2 - O(k\sqrt{n}\sqrt{\mu}), \tag{14}$$

*Proof of Lemma 11.* Recall that to obtain $\overline{D}$, we first obtain $\gamma D$ (Line 1 in Algorithm 2) and then randomly round the entries in $\gamma D$ to the nearest integers (Lines 2-6 in Algorithm 2). Hence, we can decompose $\overline{D}$ as $\gamma D + E_d$, where $E_d$ is the random matrix due to stochastic rounding. Every entry of $E_d$ is independent and is of mean 0. Hence, we can rewrite $\tilde{C}$ as follows.

$$\tilde{C} = \gamma^2 D^T D + \underbrace{2\gamma E_d^T D + E_d^T E_d + E_p}_{\text{denoted as } E}, \tag{15}$$

where $E_p$ is the symmetric random matrix, where each entry on the upper diagonal is independently sampled from $Sk(2\mu)$. We define $E := 2\gamma E_d^T D + E_d^T E_d + E_p$, the random matrix due to discretization and perturbation.

Since $\tilde{V}_k$ (not $V_k$) is the principal subspace of matrix $\tilde{C}$, we have

$$Tr(\tilde{V}_k^T(\gamma^2 D^T D + E)\tilde{V}_k) \geq Tr(V_k^T(\gamma^2 D^T D + E)V_k)$$

$$\geq Tr(\gamma^2 V_k^T D^T D V_k) + Tr(V_k E V_k)$$

$$\geq Tr(\gamma^2 V_k^T D^T D V_k) - k\|E\|_2.$$

Hence,

$$Tr(\tilde{V}_k^T(\gamma^2 D^T D)\tilde{V}_k) \geq Tr(\gamma^2 V_k^T D^T D V_k) - 2k\|E\|_2,$$

Hence,

$$Tr(\tilde{V}_k^T D^T D \tilde{V}_k) \geq Tr(V_k^T D^T D V_k) - \frac{2k}{\gamma^2}\|E\|_2,$$

which is equivalent to

$$\|D\tilde{V}_k\|_F^2 \geq \|DV_k\|_F^2 - \frac{2k}{\gamma^2}\|E\|_2. \tag{16}$$

It suffices to bound the spectral norm of $E$, which is the sum of $2\gamma E_d^T D$, $E_d^T E_d$, and $E_p$. First, note that each entry of $E_d$ is independent and has zero mean and $D$ is a deterministic matrix. Vershynin (2011) showed that the spectral norm of matrix $E_d^T D$ has mean $O(\sqrt{n})$ and variance smaller than $3n$. Applying Chebyshev's inequality, we can see that the spectral norm of matrix $E_d^T D$ is $O(n)$ with high probability. Next, for $E_d^T E_d$, we have that $\|E_d^T E_d\|_2 \leq \|E_d^T\|_2 \|E_d\|_2$, where both $E_d = E_d \mathbf{I}$ and $E_d^T = E_d^T \mathbf{I}$ have spectral norms of expectation $\sqrt{n}$. Using the same argument, we have that the spectral norm of matrix $E_d^T E_d$ is also of $O(n^2)$ with high probability. To bound the spectral norm of matrix $E_p$, we first review some basic properties of the symmetric Skellam distribution $\mathrm{Sk}(\mu)$. The distribution of $\mathrm{Sk}(\mu)$ is obtained by taking the difference of two independent Poisson random variates sampled from $\mathrm{Pois}(\mu)$. The moment generating function of $X \sim \mathrm{Sk}(\mu)$ is written as follows

$$\mathbb{E}[e^{\lambda X}] = e^{\mu(e^\lambda + e^{-\lambda} - 2)}. \tag{17}$$

In particular, for $0 \leq \lambda < 1$, we have that $e^\lambda + e^{-\lambda} - 2 \leq 1.09\lambda^2$. Hence, for $|\lambda| < 1$,

$$\mathbb{E}[e^{\lambda X}] = e^{1.09\mu\lambda^2}. \tag{18}$$

Hence, $Z \in \mathrm{SE}(2.18\mu, 1)$. Namely, $Z$ is sub-exponential with parameters $\sqrt{2.18\mu}$ and $1$, which means that the spectral norm of the $n$-by-$n$ symmetric random matrix whose entries on the upper diagonal are independently distributed as $Z$ is bounded by $\sqrt{n}$ with high probability (i.e., $1 - o(1/n)$) (Bandeira & van Handel, 2016; Dai et al., 2023). While each entry of $E_p$ is of scale roughly $\mu = O(\gamma^2)$ (hence its spectral norm), the factor of $\gamma^2$ gets canceled when dividing it by $\gamma^2$ in Eq. 16. The proof then follows from the triangle inequality. $\qquad\square$

The proof of Lemma 3 then follows from Lemma 11 and the fact that to achieve $(\epsilon, \delta)$-**server-observed** DP, it suffices to set $\mu_{\epsilon,\delta} = c^2 \log(1/\delta)/\epsilon^2$, where $c$ is some constant (Valovich & Aldà, 2017).

Similarly, we can obtain the privacy-utility trade-off for **SPCA** under the RDP framework by replacing $\sqrt{\mu_{\epsilon,\delta}}$ with $\sqrt{(\alpha+3)/\tau}$ (setting $\Delta_1 = \Delta_2^2$ in Lemma 2), formalized as follows.

**Lemma 12.** *Let $V_k$ be the rank-$k$ subspace of the original matrix $D$ and let $\tilde{V}_k$ be the principal rank-$k$ subspace of matrix $\tilde{C}$ obtained from Algorithm 1 with discretization parameter $\gamma = O(n)$. Then Algorithm 1 satisfies $(\alpha, \tau)$-**server-observed** RDP and with high probability, we have that*

$$\|D\tilde{V}_k\|_F^2 \geq \|DV_k\|_F^2 - O(k\sqrt{n(\alpha+3)/\tau}). \tag{19}$$

**Remark on the effect of $N$ on result utility.** The utility result of Lemma 3 and the empirical performance of SPCA is indeed of $N$, which is exactly why SPCA achieves comparable performance as the strong centralized baseline (think of it as $N = 1$). To explain, recall that the core idea of SPCA is to aggregate $N$ independent Skellam noises distributed as $Sk(\mu/N)$ from the client side into $Sk(\mu)$, a larger Skellam noise. Here, the scale of $\mu$, which determines the result utility, is in turn determined by the required server-observed level of DP, which is independent of $N$.

Instead, what $N$ influences is the level of client-observed DP (with the level of server-observed DP fixed), since each client knows her local DP noise out of the overall N contributions. As $N$ increases, the client-observed level of DP becomes stronger since each client's knowledge about the overall noise becomes smaller.

**Remark on the effect of $\gamma$ on communication and computational cost.** $\gamma$ represents the level of quantization granularity, which is reflected as the number of bits (namely, $\log \gamma$) in communication. For example, quantizing a real number from 0 to 1 with $\gamma = 2^{12}$ results in a 12-bit vector. In the case of the classic BGW protocol, the communication is $O(\log \gamma)$ for sharing the secrets. The time complexity, on the other hand, is independent of $\gamma$ as the computation for constructing the secret shares (matrix multiplication) and reconstructing the secret (polynomial interpolation) are done on the client side, and the algorithms for these computations are usually dependent on the size of the input, rather than the range of the input. For example, the Newton interpolation on $N$ points ($N$ secret shares of all clients) incurs a time complexity of $O(N^2)$, which is independent of $\gamma$. In general, the time and communication complexity vary for different MPC protocols and implementations.

A.4 EXPERIMENTS ON PCA

We include an illustration for $k = 2$ on KDDCUP in Figure 2, where both the centralized-DP algorithm and our solution obtain high-quality 2-dimensional subspaces for the input dataset, as the normal and not normal points are separated by a clear margin when projecting onto the subspaces. On the other hand, it is difficult to separate the points on the subspace obtained using the distributed baseline solution.

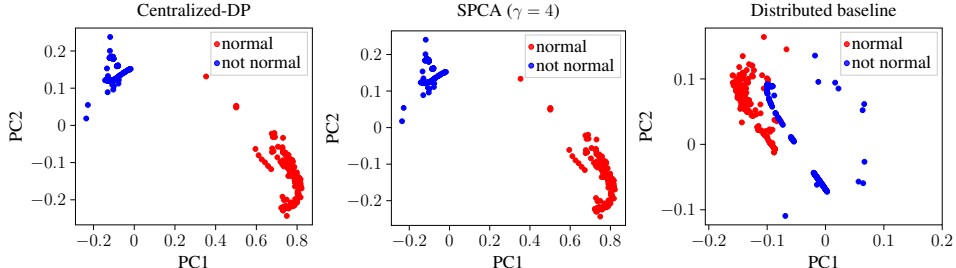

Figure 2: Visualization of the top-2 subspace computed on KDDCUP under $\epsilon = 0.25$, $\delta = 10^{-5}$.

We fix $\delta = 10^{-5}$ and report the corresponding levels of $\epsilon_{\text{server}}$ under server-observed DP and $\epsilon_{\text{client}}$ under client-observed DP of the distributed algorithms used in PCA experiments in Tables 1 and 2. Note that our solution significantly outperforms the baseline in terms of the privacy-utility trade-off, although our solution incurs higher privacy costs than the baseline in terms of client-observed DP under the same levels of server-observed DP. For example, our mechanism with $\epsilon_{\text{client}} = 1.54$ (stronger server-observed privacy) significantly outperforms the distributed baseline with $\epsilon_{\text{client}} = 2.16$ (weaker server-observed privacy) for both datasets when $\gamma$ is large enough (see Figure 1 in the main paper). Similar conclusion also applies to the other two high-dimensional datasets CiteSeer and Gene, and we omit the details here.

Table 1: Corresponding levels of $\epsilon_{\text{server}}$ (within $0.01$ error) under server-observed DP and $\epsilon_{\text{client}}$ under client-observed DP of distributed algorithms used in PCA experiments on KDDCUP dataset ($n = 816$). The $\delta$ is fixed to $10^{-5}$.

| $\epsilon_{\text{server}}$ / $\epsilon_{\text{client}}$ | 0.25 | 0.5 | 1 | 2 | 4 | 8 | 16 |
|---|---|---|---|---|---|---|---|
| **SPCA ($\gamma = 4$)** | 0.77 | 1.54 | 3.14 | 6.53 | 13.75 | 29.92 | 70.33 |
| **SPCA ($\gamma = 6$)** | 0.77 | 1.54 | 3.14 | 6.53 | 13.73 | 29.80 | 70.22 |
| **SPCA ($\gamma = 12$)** | 0.77 | 1.54 | 3.14 | 6.53 | 13.71 | 29.75 | 69.84 |
| **Baseline** | 0.54 | 1.07 | 2.16 | 4.41 | 9.09 | 19.14 | 40.89 |

Table 2: Corresponding levels of $\epsilon_{\text{server}}$ (within $0.01$ error) under server-observed DP and $\epsilon_{\text{client}}$ under client-observed DP of distributed algorithms used in PCA experiments on ACSIncome dataset ($n = 117$). The $\delta$ is fixed to $10^{-5}$.

| $\epsilon_{\text{server}}$ / $\epsilon_{\text{client}}$ | 0.25 | 0.5 | 1 | 2 | 4 | 8 | 16 |
|---|---|---|---|---|---|---|---|
| **SPCA ($\gamma = 4$)** | 0.77 | 1.54 | 3.14 | 6.53 | 13.73 | 29.91 | 70.48 |
| **SPCA ($\gamma = 6$)** | 0.77 | 1.54 | 3.14 | 6.53 | 13.73 | 29.78 | 69.90 |
| **SPCA ($\gamma = 12$)** | 0.77 | 1.54 | 3.14 | 6.53 | 13.73 | 29.77 | 69.86 |
| **Baseline** | 0.54 | 1.07 | 2.16 | 4.41 | 9.09 | 19.14 | 40.89 |

## B EXTENSION TO LOGISTIC REGRESSION

In this section, we extend our idea of SPCA to solve the problem of DP logistic regression (LR) under Vertical FL. Given weights $w \in \mathbb{R}^n$, the logistic regression model predicts label $\hat{y} = 1$ for a feature vector $a \in \mathbb{R}^n$ with probability $\Pr[\hat{y} = 1] = \sigma(\langle w, a \rangle)$, and predicts label $\hat{y} = 0$, otherwise.

Here $\sigma(u) := 1/(1 + \exp(-u))$, is called the sigmoid function. For every record in $D$, we regard its features and label together as a concatenated vector, written as $(x, y) \in \mathbb{R}^{n+1}$, which is partitioned by $N$ clients. Here our notion of privacy protects the entire $(x, y)$, including *both* the feature information and the labels information. Without loss of generality, we assume that $\|x\|_2 \leq 1$.

## B.1 Logistic Regression with Gradient Descent

The server is interested in minimizing the cross-entropy loss averaged over $D$. Namely, he tries to solve

$$\arg\min_w \frac{1}{|D|} \sum_{(x,y) \in D} L(\sigma(\langle w, x \rangle), y), \text{ where} \tag{20}$$

$$L(\sigma(\langle w, x \rangle), y) = -y \log(\sigma(\langle w, x \rangle)) - (1 - y) \log(1 - \sigma(\langle w, x \rangle)).$$

We focus on using the gradient descent algorithm Kiefer & Wolfowitz (1952); Robbins & Monro (1951); Bottou et al. (2018) to solve Eq. 20. The gradient descent algorithm is widely used in federated learning McDonald et al. (2010); Dean et al. (2012); Coates et al. (2013); Abadi et al. (2016a). We refer interested readers to Boyd & Vandenberghe (2004); Kochenderfer & Wheeler (2019) for other optimization algorithms. The high-level idea of gradient descent is to repeatedly update $\mathbf{w}$ towards the opposite direction of the loss function's gradient $\sum_{(x,y) \in D} g(x, y)$, where $g(x, y)$ denotes the gradient for a single record $(x, y)$. The computation for the $j$-th $(j = 1, \ldots, n)$ dimension of $g(x, y)$ is as follows

$$g(x, y)_j = \sigma(\langle w, x \rangle) \cdot x_j - y \cdot x_j. \tag{21}$$

Next, we show how to optimize Eq. 20 through gradient descent, while achieving both server-observed DP and client-observed DP.

## B.2 Approximating the Gradient

As we can see from Eq. 21, the computation of $\mathbf{g}(\mathbf{x}, y)_j$ cannot be decomposed as subroutines of additive and multiplicative operations, due to the presence of the sigmoid function. As a result, the clients cannot utilize secret sharing (such as BGW protocol) as well as other MPC protocols to compute the outcome. However, recall that the sigmoid function can be approximated by polynomials using Taylor series: $\sigma(u) = \sum_{h=0}^{\infty} \left( \frac{\sigma^{(h)}(u)|_{u=0}}{h!} \cdot u^h \right)$, where $\sigma^{(h)}(u)|_{u=0}$ represents the $h$-th order derivative of $\sigma(u)$ evaluated at $u = 0$. In this work, we consider $H = 1$ (extending to high orders is not the focus of this work), then we can write $\sigma(u) \approx \frac{1}{2} + \frac{1}{4} \cdot u$. By plugging in this approximation of $\sigma(u)$ into Eq. 21, we can approximate the $j$-th dimension of the gradient $g(x, y)_j$ with

$$g(x, y)_j \approx \frac{1}{2} \cdot x_j + \frac{1}{4} \cdot (\langle w, x \rangle) \cdot x_j - y \cdot x_j, \tag{22}$$

which is a polynomial of the input that can be computed with differential privacy using secret sharing and additive DP noise.

We introduce the overall algorithm called SLR, described as in Algorthm 4. At the beginning of every iteration, the server first discretizes the model weights $w$ using Algorithm 2 and shares the discretized version $\overline{w}$ to all the clients (Line 2). Next, the clients use shared randomness to sample a record $(x, y)$ from $D$ with some probability $b$, by flipping a random coin (Line 5). Note that the subsampling procedure for a record from $D$ has to be done by the clients using shared randomness, otherwise it is not guaranteed that the sampled outcomes by different clients correspond to a single valid record. Similar to what we did in SPCA, for every sampled record $(x, y)$, the clients then *independently* and *privately* discretize every dimension of the record, using Algorithm 2, and places the discretized record $(\overline{x}, y)$ into batch $B$.

Before proceeding to the actual computation, we introduce some notations. Given a set of discretized records $B = \{(\overline{x}, \overline{y})\}$ and discretized model parameters $\overline{w}$. We define the $j$-th dimension ($j =$

---

**Algorithm 4:** SLR

---

**Input:** Initial model weights $w \in \mathbb{R}^n$; private dataset $D$ partitioned by $N$ clients; number of training
 iterations $T$; sampling parameter $b$; discretization parameter $\gamma$; noise parameters $\mu(1)$, $\mu(2)$, and
 $\mu(3)$; learning rate $\eta$; clipping norm for model weights $c_w$.

1 **for** $t = 1 \ldots T$ **do**
2    $\overline{w} \leftarrow$ Algorithm 2$(w, \gamma)$. // the server discretizes model weights $w$, and shares $\overline{w}$ to all clients.
3    $B \leftarrow \emptyset$ // initialize the batch for one iteration.
4    **for** $(x, y) \in D$ **do**
5      Flip a coin with heads probability $b$.
6      **if** *Heads* **then**
7        $(\overline{x}, \overline{y}) \leftarrow$ Algorithm 2$((x, y), \gamma)$ // the clients discretize the sampled record.
8        $B \leftarrow B \cup \{(\overline{x}, \overline{y})\}$. // place the discretized record into $B$.
9    **for** $h = 1, 2, 3$ **do**
10      Clients perturb each dimension of $(\Phi(B, h))$ with $\mathrm{Sk}(\mu)$ using the BGW protocol and send the
 outcome to the server
11    The server reconstructs $\tilde{g}(B)$ from $\{\tilde{\Phi}(B, h)\}_{h=1,2,3}$ as in Eq. 26.
12 $w \leftarrow w - \eta \cdot \tilde{g}(B)$. // server updates model parameters (Adam or SGD does not affect privacy).
13 $w \leftarrow \min(1, \frac{c_w}{\|w\|_2}) \cdot w$. // clip model weights by $\mathcal{L}_2$ norm.
**Output:** **w** model weights learnt on $D$.

---

$1, \ldots, n$) for functions $\{\Phi(B, h)\}_{h=1,2,3}$ as follows.

$$\Phi(B, 1)_j = \sum_{(\overline{x}, \overline{y}) \in B} \overline{x}_j, \tag{23}$$

$$\Phi(B, 2)_j = \sum_{(\overline{x}, \overline{y}) \in B} (\langle \overline{w}, \overline{x} \rangle) \cdot \overline{x}_j, \text{ and} \tag{24}$$

$$\Phi(B, 3)_j = \sum_{(\overline{x}, \overline{y}) \in B} \overline{y} \cdot \overline{x}_j. \tag{25}$$

The clients then compute perturbed versions for Eq. 23, Eq. 24 and Eq. 25 with additive Skellam
noises using the BGW protocol (each client contributes to $1/N$ proportion of the overall Skellam
noise $\mathrm{Sk}(\mu)$), and send the outcomes to the server (Lines 9 and 10).

Finally, the server obtains an estimate for the gradient sum by post-processing (Line 11). In particular,
the server computes

$$\tilde{g}(B) = \frac{1}{2\gamma} \cdot \tilde{\Phi}(B, 1) + \frac{1}{4\gamma^3} \cdot \tilde{\Phi}(B, 2) - \frac{1}{\gamma^2} \tilde{\Phi}(B, 3). \tag{26}$$

Here the powers of $\gamma$ are used to down-scale the outcomes computed on the discretized inputs, similar
to what we did for PCA. After the reconstruction is done, the server updates the model weights using
the estimated gradient (Line 12), and clips the model weights by $\mathcal{L}_2$ norm (Line 13). Note that the
server could use any update algorithm since post-processing preserves DP.

### B.3 PRIVACY ANALYSIS

We present the privacy guarantees of SLR. For discretization parameter $\gamma$; model dimension $n$;
and model clipping norm $c_w$, we define $\Delta_2(1) = \sqrt{\gamma^2 + n}$, $\Delta_2(2) = \sqrt{\gamma^2 c_w^2 + n} \cdot (\gamma^2 + n)$, and
$\Delta_2(3) = \gamma \sqrt{\gamma^2 + n}$. For every $h = 1, 2, 3$, we also define $\Delta_1(h)$ as the smaller one of $\Delta_2^2(h)$ and
$\sqrt{n} \cdot \Delta_2(h)$, and denote the noise parameters as $\mu(1)$, $\mu(2)$, and $\mu(3)$. Then SSS-LR satisfies both
server-observed DP and client-observed DP.

**Lemma 13.** *Running Algorithm 4 for $T$ iterations satisfies $(\alpha, \tau_{server})$-server-observed RDP with*

$$\tau_{server} = \frac{T}{\alpha - 1} \log \left( (1 - b)^{\alpha - 1}(\alpha b - b + 1) + \sum_{l=2}^{\alpha} \binom{\alpha}{l}(1 - b)^{\alpha - l} b^l e^{(l-1)\tau_l} \right),$$

Table 3: Corresponding levels of $\epsilon_{\text{server}}$ under **server-observed** DP and $\epsilon_{\text{client}}$ under **client-observed** DP of distributed algorithms used in LR experiments on all datasets. The $\delta$ is fixed to $10^{-5}$.

| $\epsilon_{\text{server}}$ / $\epsilon_{\text{client}}$ | 0.5 | 1 | 2 | 4 | 8 | 16 |
|---|---|---|---|---|---|---|
| **SLR** ($\gamma = 12, 14, 16$) | 24.62 | 48.84 | 90.02 | 133.33 | 329.28 | 464.67 |
| **Baseline** | 1.07 | 2.16 | 4.41 | 9.09 | 19.14 | 40.89 |

*where $\tau_l$ is computed as follows*

$$\tau_l = \sum_h \left( \frac{l\Delta_2^2(h)}{4\mu(h)} + \min\left( \frac{(2l-1)\Delta_2^2(h) + 6\Delta_1(h)}{16\mu^2(h)}, \frac{3\Delta_1(h)}{4\mu(h)} \right) \right).$$

**Lemma 14.** *Let $\beta = \frac{N}{N-1}$, and $T_{sample}$ be the number of times that a record $(\mathbf{x}, y) \in D$ is sampled into the batch $B$, then for that particular record, Algorithm 4 satisfies $(\alpha, \tau_{client})$-client-observed RDP with*

$$\tau_{client} = T_{sample} \cdot \left( \sum_{h=1,2,3} \left( \frac{\alpha\beta\Delta_2^2(h)}{\mu(h)} + \min\left( \frac{(2\alpha-1)\beta^2\Delta_2^2(h) + 3\beta^2\Delta_1(h)}{4\mu^2(h)}, \frac{3\beta\Delta_1(h)}{2\mu(h)} \right) \right) \right).$$

### B.4 EXPERIMENTS ON LOGISTIC REGRESSION

We evaluate LR on the ACSIncome and ACSEmployment datasets collected from four states of the US: California, Texas, New York, and Florida (Ding et al., 2021). The tasks are predicting whether a person has an annual income over $50K$ and whether a person is employed, for the ACSIncome and ACSEmployment datasets, respectively. We compare the performance of SLR versus the baseline VFL solution, which first perturbs the dataset $D$ on the client side and then trains the LR model on the perturbed dataset. The baseline solution of DP LR for vertically partitioned data is essentially the same as that for PCA. Namely, each client independently perturbs her local data partition before sharing it with one of the clients (or the server), who then performs LR on the perturbed dataset. In addition, we use the ubiquitous centralized-DP mechanism, DPSGD, as the strong baseline. We omit other centralized-DP mechanisms since they perform similarly to DPSGD, and our goal is not to evaluate DP mechanisms for the centralized setting. We also vary the discretization parameter $\gamma$ from $\{12, 14, 16\}$ while fixing the model clipping norm $c_w = 1$. We do not tune the hyperparameters in favor of any algorithm.

For all experiments, we fix the privacy parameter $\delta = 10^{-5}$ and vary $\epsilon$ from $\{0.5, 1, 2, 4, 8, 16\}$. Similar to PCA, the privacy level is calculated based on **server-observed** privacy, and the corresponding **client-observed** levels of privacy are listed in Table 3. We report the average test accuracy over 10 independent runs in Figure 3. From Figure 3, we can see that SLR significantly outperforms the baseline solution under all parameter settings for all datasets. In particular, when $\epsilon$ is small, the utility of the baseline solution is below the lower limit shown in the figures while our solution can still maintain a decent utility. In addition, given a sufficient amount of privacy budget, SLR achieves comparable performance as the strong centralized-DP baseline. For example, considering the ACSIncome (CA) and ACSIncome (TX) datasets, the accuracy drop of our solution with $\gamma = 16$ is close to the strong baseline, with a small accuracy drop of 2% to 3% when $\epsilon \geq 4$. For the ACSEmployment (CA), ACSEmployment (FL), and ACSEmployment (TX) datasets, the accuracy drop of our solution for all choices of $\gamma$ from the strong baseline is 1% to 2% when $\epsilon \geq 4$.

In terms of hyperparameters, for $\epsilon = 0.5, 1, 2, 4, 8,$ and $16$, we train the LR model for $5, 5, 8, 10, 20,$ and 20 epochs, respectively, for all datasets and algorithms. We fix the subsampling rate for every iteration to 0.001. In SLR, we also balance the relative errors due to DP noise in all three functions of $\{\Phi(B, h)\}_{h=1,2,3}$. To do this, we consider a notion of noise multiplier $\mu_{\text{multi}}$, which was also used in the DPSGD literature Abadi et al. (2016b). For $h = 1, 2,$ and $3$, we set $\mu(h)$ to be $\gamma^2 \cdot \mu_{\text{multi}}$, $\gamma^6 \cdot \mu_{\text{multi}}$, and $\gamma^4 \cdot \mu_{\text{multi}}$, respectively, as the server needs to downscale $\{\Phi(B, h)\}_{h=1,2,3}$ by $\gamma, \gamma^3$, and $\gamma^2$, respectively. The value of $\mu_{\text{multi}}$ is then computed from Lemma 13.

Similar to PCA, we fix $\delta = 10^{-5}$ and report the corresponding levels of $\epsilon_{\text{server}}$ under **server-observed** DP and $\epsilon_{\text{client}}$ under **client-observed** DP of distributed algorithms used in LR experiments for all

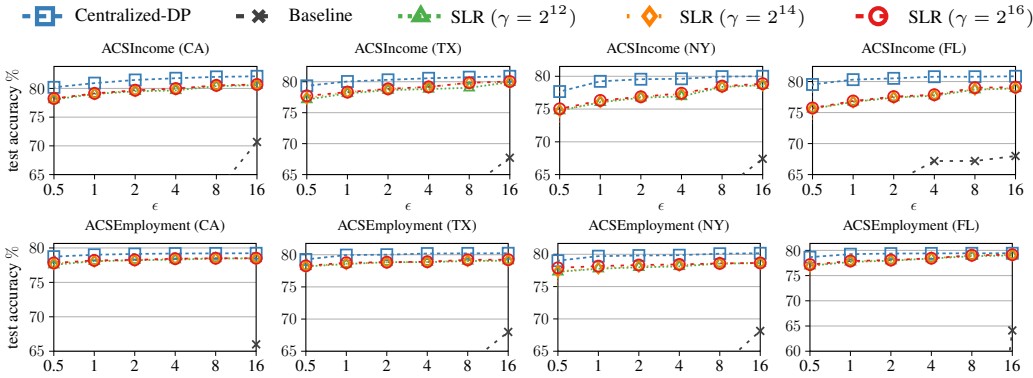

Figure 3: Performance for logistic regression on US census data with $\delta$ fixed to $10^{-5}$ while varying $\epsilon$.

datasets in Table 3. Under the same level of server-observed DP, our solution incurs higher privacy costs than the baseline in terms of client-observed DP. This is because the baseline solution needs to release the sensitive information only once while our solution is based on gradient descent, a composite algorithm. Further reducing the client-level privacy cost is a promising future work direction. Similar to PCA, we would like to emphasize that our solution significantly outperforms the baseline under the same level of server-observed DP. For example, our mechanism with $\epsilon_{\text{client}} = 24.62$ (stronger server-observed privacy) significantly outperforms the distributed baseline with $\epsilon_{\text{client}} = 40.89$ (weaker server-observed privacy) for all datasets, regardless of the choices of $\gamma$. These observations confirm the efficacy of our solution under both server-observed and client-observed DP frameworks for both tasks of PCA and LR.

