# OpenReview forum: "Differentially Private Principal Component Analysis for Vertically Partitioned Data"
_ICLR.cc/2024/Conference — Submitted to ICLR 2024_

### Official Review · Reviewer_qWzN · 2023-10-28

**Soundness:** 3 good
**Presentation:** 3 good
**Contribution:** 1 poor
**Rating:** 1
**Confidence:** 5

**Summary:**

This research paper provides a comprehensive solution called Secure Principal Component Analysis (SPCA) for differentially private principal component analysis (PCA) in vertical Federated Learning. SPCA technique introduces minimal noise to the obtained subspace while preserving differential privacy (DP) without assuming any trusted client or third party. Ensuring data privacy can be a challenge, especially when clients are adversarial. SPCA, on the other hand, ensures privacy protection for both the server and clients. The authors provide a theoretical analysis that indicates that it can achieve the same level of privacy-utility trade-off as the optimal baseline in a centralized setting. Through experiments on real-world datasets, the researchers demonstrate that SPCA achieves optimal error rates comparable to the centralized baseline. Overall, this paper presents a new solution for DP-PCA on vertically partitioned data, with a theoretical analysis demonstrating its effectiveness.

**Strengths:**

1) The paper proposes a solution called Secure Principal Component Analysis (SPCA) for differentially private principal component analysis in vertical Federated Learning.
2) The paper presents a theoretical analysis demonstrating that SPCA can achieve the privacy-utility trade-off of the optimal baseline in the centralized setting. The analysis shows the solution's effectiveness.
3) The research paper presents real-world experiments that confirm the theoretical analysis. The paper demonstrates how the analysis has been validated on various datasets.

**Weaknesses:**

1) Although the paper includes experiments on real-world datasets, the number of experiments is relatively small, which could limit the generalizability of the results.

2) Comparisons with other existing solutions for differentially private principal component analysis for vertically partitioned data are not presented in the paper.

**Questions:**

1) Wang et al. did "Differentially Private Principal Component Analysis Over Horizontally Partitioned Data," but what is your novelty for vertically partitioned data?
2) Can you elaborate on how the proposed solution can be practically applied?

---

> ### Author Response · Authors · 2023-11-13
> **Response to Reviewer qWzN**
>
> Thanks for the comments. In what follows, we provide a detailed response. Please kindly let us know if you have any further questions.
>
> ---
>
> **Weaknesses 1. Although the paper includes experiments on real-world datasets, the number of experiments is relatively small, which could limit the generalizability of the results.**
>
> ---
>
> **Response.** We believe this is a misunderstanding. First of all, we have included an *extensive* set of experiments, covering different parameter regimes, including both high-dimensional datasets with $n\ll m$ (CiteSeer and Gene), as well as the KDDCUP dataset, which was commonly used for evaluating PCA (e.g., see Chaudhuri et al. [1]), and the recently released ACSIncome dataset [2] that is commonly used for evaluating FL algorithms. In terms of *theory*, we have also proved that our solution is able to achieve comparable performance as the strong centralized baseline.  Both the empirical results and theoretical analysis confirm the performance and generalizability of our solution.
>
> ---
>
> **Weaknesses 2. Comparisons with other existing solutions for differentially private principal component analysis for vertically partitioned data are not presented in the paper.**
>
> ---
>
> **Response.** Thanks for raising this concern. As we have mentioned, existing solutions for private PCA for vertically partitioned data that utilize MPC alone do not provide any rigorous DP guarantees (e.g., see [3,4]). Other works, which attempted to achieve DP under vertical FL by letting one of the clients or a third party to inject random DP noises to the sensitive outcome, are **not differentially private** either, as the party might be curious and infer the sensitive outcome (e.g., see [5,6]). As a result, these baselines are **not comparable** with ours, which simultaneously preserves two levels of DP, regarding both a curious server and the curious clients in FL, without assuming any trusted third parties. The only baseline that is comparable to ours is the baseline introduced on page 6 (detailed in Algorithm 3 of page 14), whose performance is much worse than our SPCA, as illustrated in both the theoretical analysis and empirical evaluations.

---

> ### Author Response · Authors · 2023-11-13
> **Response to Reviewr qWzN (continued)**
>
> **Question 1. Wang et al. did "Differentially Private Principal Component Analysis Over Horizontally Partitioned Data," but what is your novelty for vertically partitioned data?**
>
> ---
>
> **Response.** Thanks for the question. We would like to first clarify that existing DP solutions for **horizontally partitioned data** (including the approach by Wang et al. [7]) **do not apply** to our setting of **vertically partitioned data**. The key reason, as we have mentioned in the introduction section, is non-linearity. In the horizontal setting, the clients are able to linearly aggregate their locally perturbed results whereas in the vertical setting, different clients must collaborate to compute the target. In the case of PCA, each entry of the covariance matrix is computed as the inner product between two column vectors possessed by different clients. As a result, the perturbation process cannot be done by a single client alone, since otherwise, the client would learn the sensitive outcome, violating differential privacy.
>
> In terms of contribution, we present the first DP algorithm for PCA over *vertically partitioned data* that simultaneously protects the partitioned data from both the curious server and curious clients, without assuming any trusted third parties. In addition, our algorithm achieves comparable performance as the strong centralized baseline, which is also **a first of its kind** in the vertical FL literature. Although it is well known that DP algorithms under *horizontal FL* are able to achieve comparable performance as in the centralized setting, with the help of secure aggregation or secure shuffling, whether the same can be done under *vertical FL*, however, *has not been answered until this work*. We provide a positive answer in this work, which, we believe is a notable contribution to the privacy community.
>
> ---
>
> **Question 2. Can you elaborate on how the proposed solution can be practically applied?**
>
> ---
>
> **Response.** To apply our solution in practice, there are four steps in general, as outlined in Algorithm 1 on page 7.
>
> First, each client independently discretizes her local data partition. This computation can be done efficiently on the client side since only multiplication and random rounding are involved (see Algorithm 2).
>
> Next, each client independently samples Skellam noises on the local side privately, which is done by taking the difference between two identically distributed independent Poisson random variables. Sampling algorithms for Poisson are implemented in commonly used libraries such as NumPy and SciPy.
>
> The clients then collaboratively compute the sum of the covariance matrix and the sampled local Skellam noises (as in Eq.(7) on page 6) using any secure multiparty computation (MPC) algorithm without affecting the privacy guarantees (e.g., using the classic BGW protocol). The computation of Eq.(7) involves only summation and multiplication, which are considered elementary operations and can be done efficiently in modern MPC algorithms (e.g., SPDZ by Keller et al. [8]).
>
> Finally, the server reconstructs the covariance matrix from the outcome of MPC in the previous step and computes the $k$-dimensional singular subspace from the covariance matrix on her side. The server is free to use any centralized algorithm for the computation without affecting the privacy guarantees since post-processing preserves DP (e.g., using the power iteration method for efficient computation of eigenvectors [9]).

---

> > ### Author Response · Authors · 2023-11-13
> > **References.**
> >
> > **References.**
> > > [1]. K. Chaudhuri, A. Sarwate, and K. Sinha. A near-optimal algorithm for differentially-private principal components. JMLR, 14(1):2905–2943, 2013.
> >
> > > [2]. Frances Ding, Moritz Hardt, John Miller, and Ludwig Schmidt. 2021. Retiring Adult: New Datasets for Fair Machine Learning. In NeurIPS. 6478–6490.
> >
> > > [3]. Yiu-ming Cheung and Feng Yu. Federated-pca on vertical-partitioned data. 05 2020. doi: 10.36227/techrxiv.12331649.v1.
> >
> > > [4]. Xiaoyu Fan, Guosai Wang, Kung Chen, Xu He, and Weijiang Xu. Ppca: Privacy-preserving principal
> > component analysis using secure multiparty computation(mpc). ArXiv, abs/2105.07612, 2021.
> >
> > > [5]. Depeng Xu, Shuhan Yuan, and Xintao Wu. Achieving differential privacy in vertically partitioned
> > multiparty learning. In Big Data, pp. 5474–5483, 2021.
> >
> > > [6]. Thilina Ranbaduge and Ming Ding. Differentially private vertical federated learning. CoRR,
> > abs/2211.06782, 2022.
> >
> > > [7]. Sen Wang and J. Morris Chang. Differentially private principal component analysis over horizontally
> > partitioned data. In 2018 IEEE Conference on Dependable and Secure Computing (DSC), pp. 1–8,
> > 2018. doi: 10.1109/DESEC.2018.8625131.
> >
> > > [8]. Marcel Keller, Valerio Pastro, and Dragos Rotaru. Overdrive: Making spdz great again. In Jesper Buus
> > Nielsen and Vincent Rijmen (eds.), EUROCRYPT 2018, pp. 158–189, 2018.
> >
> > > [9]. Gene H. Golub, Henk A. van der Vorst. Eigenvalue computation in the 20th century. In Journal of Computational and Applied Mathematics, Volume 123, Issues 1–2,
> > 2000, Pages 35-65.

---

> ### Author Response · Authors · 2023-11-19
> **Any further questions or comments?**
>
> We want to thank the reviewer for their comments.
>
> As the deadline for the rebuttal phase is approaching, we would like to encourage the reviewer to engage in the interactive rebuttal to help improve our paper and clarify misunderstandings.

---

### Official Review · Reviewer_fmuB · 2023-10-30

**Soundness:** 3 good
**Presentation:** 3 good
**Contribution:** 2 fair
**Rating:** 6
**Confidence:** 5

**Summary:**

This paper proposes a differentially private technique for PCA on vertically partitioned data.

**Strengths:**

The approach is mostly sound (but see a few important notes below).
The text is well written and understandable.
Probably the most important contribution is the analysis of the privacy as a function of noise (if it is correct).

**Weaknesses:**

The paper mentions secret sharing, but then doesn't mention an important baseline to consider.  In particular, the brute-force approach which needs "minimal (*)" noise is that the clients secret-share their portion of the data, that they compute the covariance matrix together in secret shared form (this only requires additions and multiplications, which under SSS can be performed rather efficiently once the data is secret shared), together add the (minimally required) noise, and then reveal the noisy covariance matrix C.  This algorithm is clearly differentially private and doesn't leak intermediate results.  The only thing another approach can hope to do better is to require fewer communication (and computation).  Unless you can show that your proposal is significantly less expensive than this fully secret-sharing based approach, it doesn't seem a very valuable contribution.  In fact, it seems that Algorithm 1 is not much more efficient that the brute force algorithm I sketch above as it needs to involve all N clients for the computation of each inner product of columns i and j not belonging to the same client.  I guess Algorithm 1 isn't significantly faster than the baseline I describe above, while you could have made it faster by computing D[:,i].D[:,j] using a multi party computation involving only the clients owning columns i and j.

While \sum_{q=1}^N z_q offers some privacy, every client k knows z_k and can therefore compute \sum_{q=1}^{k-1} z_q + \sum_{k+1}^N z_q which is the sum of only (N-1) noise terms and hence gives only (N-1)/N of the privacy provided by Sk(\mu).  Ideally, in line 4 of Algorithm 1 clients should sample from Sk(\mu/(N-1)).  Alternatively, instead of letting all clients sample from Sk(\mu/(N-1)) it would be even better to let the clients collaborative sample from Sk(\mu) without any client learning the sampled value (i.e., sampling using SSS).  In that case, the "minimal (*)" amount of noise Sk(\mu) would be added to the final result.

When the abstract (and my comments above) say "minimal noise", this is not really the minimal noise, but the smallest amount of noise for which the proof of DP is easy and straightforward.  There is no proof that there doesn't exist an even smaller amount of noise (where in particular possibly not every component C[i,j] gets independent noise) which leads to a result which can also be proven to be DP.

The proof of the main result (privacy) contains several mistakes, and it is therefore hard to verify its overall correctness, even if I believe that at a high level the result is plausible (i.e., I'm confident such a result is possible but I don't know whether the lower-order terms or constant factors are correct).  For example:
* "Since the L2 norm for each row in D and D' is bounded by \sqrt{\gamma^2+n} ... we have that ... \|D^\top D - D'^\top D'\|_F^2 \le \gamma^2 + n" : I would expect that the bound on the norm of these inner products is also linear in \sqrt{m} with m the number of rows.
* "In addition, that the L1 norm of an integer-valued vector v is always less than or equal to \|v\|_2^2 and \sqrt{n}\|v\|_2" : this sentence isn't fully grammatically clear.  It is not correct that the L1 norm of v, i.e., \sum_i |v_i| is always smaller than \|v\|_2^2 = \sum_i v_i^2 (especially not for "integer valued v" where it is possible some components of v are larger than 1 in absolute value.
* Next, the text just calls for lemma 1, but it would help significantly if the text would first make all parameters of lemma 1 explicit, e.g., \Delta_1, \Delta_2, ...

**Questions:**

I assume that what you describe in "Baseline in vertical FL." corresponds to what is more commonly known as "Local differential privacy", i.e., every client adds so much noise that the publication of the data doesn't allow an adversary to reveal any sensitive information ?

**Details Of Ethics Concerns:**

--

---

> ### Author Response · Authors · 2023-11-13
> **Response to Reviewer fmuB**
>
> We would like to thank the reviewer for the thoughtful comments. In what follows, we provide a detailed response. Please kindly let us know if you have any further questions.
>
> ---
>
>
> **1. The brute-force baseline.**
>
> ---
>
> **Response.** Indeed, we have considered the brute-force algorithm you mention above, where for each $C(i,j)$, only the corresponding clients who own columns $i$ and $j$ (say client $p$ and client $q$) participate in the secret sharing process. However, this brute-force approach has the drawback that both client $p$ and client $q$ know exactly *half* the overall DP noise. As a result, it is *easier* for client $q$ to infer the data of client $p$, compared with our solution, where both client $p$ and $q$ know only $1/N$ of the overall DP noise. In cases when $N>2$, our solution is *more private considering curious clients*.
>
> ---
>
>
> **2. Client's knowledge of $Sk(\mu/N)$.**
>
> ---
>
> **Response.** You are correct. From a client’s perspective, the DP guarantee is provided by $Sk((N-1)\mu/N)$. Hence, the levels of client-observed DP and server-observed DP (which is provided by $Sk(\mu)$) are different, as we have stated in Lemma 2 (see Eq.(9) and Eq.(10) in page 8). For better comparison with previous works from the centralized setting and the horizontal FL setting [1,2], we focus on server-observed DP. The corresponding levels of client-observed DP are reported in Tables 1 and 2 of page 17.
>
>
> ---
>
>
> **3. The minimal noise.**
>
> ---
>
> **Response.** You are correct. We could not technically prove that our noise is minimal. We have revised the term *minimal noise* to ``noise of scale comparable with the strong centralized baseline'' in the draft.
>
> ---
>
>
> **4. Alternative solution for collaborative sampling from $Sk(\mu)$.**
>
> **Response.** Although it is a tempting idea to let the clients sample from $Sk(\mu)$ directly while getting rid of the slackness of $1/N$, we are not aware of any *efficient* and *rigorous* approach to do so under distributed FL settings (either vertical or horizontal).
>
> In particular, the slackness of $1/N$ was also observed in the state-of-the-art approaches of horizontal FL [1,2], where each client independently injects locally generated DP noises to the sensitive outcome. Wang et al. [3] attempted to remove the slackness by letting the clients jointly sample from a discrete Gaussian distribution using a shared random seed. However, as pointed out by Kairouz et al. [1], a single curious client could learn the overall DP noise from the shared seed, leading to **privacy violation**.
>
> In vertical FL, Wu et al. [3] propose to use a secret-shared random seed to generate Laplace noise to achieve DP when generating decision trees under vertical FL. However, they *did not* provide any **formal DP analysis** from the perspective of a curious client. In addition, the process for converting shared randomness to the designated distribution of DP noise (e.g., discrete Gaussian [1] and Skellam [2]) might be a huge **overhead** for the clients to bear, considering that even the centralized algorithms for generating discrete Gaussian and Skellam distributions from random bits are *complicated and time-consuming* (see Section 5 in [4], page 487 of [5], and [6]) and the clients also need to *synchronize repeatedly* in such complicated algorithms under the MPC model.
>
> On the contrary, the overhead of DP in SPCA comes from the addition of $N$ Skellam noises, as the client can generate Skellam noises in an *offline manner* without collaboration. This overhead of $N$ additions is *not significant* when compared with the computation of an element in the covariance matrix that involves the addition of $m$ multiplications, considering that the number of clients $N$ is usually much smaller than the number of records $m$ in vertical FL. In addition, when $N$ is large, the difference between $N$ and $N-1$ *becomes negligible*. Further closing this gap would be an interesting direction to explore.

---

> > ### Comment · Reviewer_fmuB · 2023-11-13
> >
> > Thanks for your clarifications.
> >
> > You don't answer my question on whether the proposed algorithm is really better than what I called the "brute force" algorithm.
> >
> > On the other hand, there is an interesting discussion on whether computing an inner product of two columns is more efficient with two or with all partners.  It seems to come down to the question whether the parties know the noise they generate.  I agree that if the parties know the noise they generate and you consider client-observed DP, then it is better to let all parties participate in the inner product computation.  However, in your response you say you mainly consider server-observed DP and are not so interested in client-observed DP.  Moreover, it is not needed that clients know the noise, and in that case the provided argument (that clients know half of the noise) isn't relevant.
> >
> > The paper and the answer about the knowledge by the clients of $Sk(\mu/N)$ suggests that the clients always generate each a part of the noise and then combine the noise.  This is a classic strategy, also used by Dwork in the early DP papers, and as you point out has been used since then in a lot of the state of the art with many authors not considering in depth better alternatives.  However, given that you anyway use secret sharing, it is possible to generate noise which none of both clients know, at a constant cost per pair of parties.  While I agree this cost may be non-trivial (you claim it is large), it is only a constant cost per pair of clients. A constant cost between each pair of parties, independent of the dataset size, may still be acceptable as nowadays datasets have become very large and any algorithm running over the complete dataset  typically has much higher cost than the joint generation of a random number.
> >
> > In conclusion, while there may be arguments why you prefer to not go beyond the state of the art in terms of oblivious noise generation, and there may be reasons to stay with an approach where all parties are involved in all inner product computations, this also makes it harder to see how exactly the proposed algorithm goes beyond a brute-force secret sharing of the PCA algorithm.
> >
> > Your resolution of point (3) seems appropriate.

---

> ### Author Response · Authors · 2023-11-13
> **Response to Reviewer fmuB (continued)**
>
> **5. Correctness about proofs.**
>
> ---
>
> **Response.** We apologize for the unclear presentation of the proofs. We have revised the proof. In particular, we have included more details to explain that the sensitivity of the covariance matrix of the quantized data is independent of $m$. We also note that since every non-negative integer $u$ is smaller than or equal to $u^2$ (equality holds for $u=0,1$), we have that $||v||_1\leq ||v||_2^2$ for any integer vector $v$. Finally, we have explicitly stated the parameters in the proof for Lemma 2. Please refer to Page 15 of the revised draft for more details.
>
> ---
>
>
> **6. I assume that what you describe in ''Baseline in vertical FL'' corresponds to what is more commonly known as ''Local differential privacy'', i.e., every client adds so much noise that the publication of the data doesn't allow an adversary to reveal any sensitive information ?.**
>
> ---
>
> **Response.** You are correct. The vertical FL baseline is indeed a local DP algorithm. As we have mentioned in the draft, other existing vertical FL approaches either involve a client, the server, or a third party to perform the noise injection step, which is not differentially private considering that these parties can be curious or do not provide any rigorous DP analysis (such as [7]).
>
>
> ---
>
> **References.**
> > [1]. Peter Kairouz, Ziyu Liu, and Thomas Steinke. 2021. The Distributed Discrete Gaussian Mechanism for Federated Learning with Secure Aggregation. In ICML.
>
> > [2]. Naman Agarwal, Peter Kairouz, and Ziyu Liu. 2021. The Skellam Mechanism for Diferentially Private Federated Learning. In NeurIPS.
>
> > [3]. Yuncheng Wu, Shaofeng Cai, Xiaokui Xiao, Gang Chen, and Beng Chin Ooi. Privacy preserving vertical federated learning for tree-based models. In PVLDB.
>
> > [4]. Clément L. Canonne, Gautam Kamath, and Thomas Steinke. 2020. The Discrete Gaussian for Diferential Privacy. In NeurIPS.
>
> > [5]. Luc Devroye. 1986. Non-Uniform Random Variate Generation. Springer. https://doi.org/10.1007/978-1-4613-8643-8
>
> > [6]. Philippe Duchon and Romaric Duvignau. 2016. Preserving the Number of Cycles of Length k in a Growing Uniform Permutation. Electron. J. Comb. 23 (2016), P4.22.
>
> > [7]. Thilina Ranbaduge and Ming Ding. Differentially private vertical federated learning. CoRR, abs/2211.06782, 2022.

---

> ### Author Response · Authors · 2023-11-15
> **Response to Reviewer fmuB**
>
> Thanks for your kind explanations.
>
> We agree that the brute-force approach you have suggested *indeed is better* than our proposed solution in terms of the privacy-utility trade-off from the perspective of a *curious client*, who no longer has the knowledge of $Sk(\mu/N)$.
>
> However, we would like to point out that, still, the difference *may be negligible* in scenarios where $N$ is large (e.g., $N=20$). In addition, in certain scenarios, the cost of the brute-force approach could be more significant (despite that it is a constant regarding the size of data). For example, considering the Skellam noise generation with a large $\mu$ (corresponding to strong privacy constraints), the clients would need to collaboratively generate $\mu$ *independent copies* of $Sk(1)$ (let's say $\mu$ is an integer for the ease of discussion), which could be time-consuming. On the other hand, in our solution, each of the $N$ clients independently generates a much smaller Skellam noise in a more efficient offline manner, which could be an advantage under strong privacy constraints and when the number of records is small.

---

### Official Review · Reviewer_DSCS · 2023-10-31

**Soundness:** 3 good
**Presentation:** 3 good
**Contribution:** 3 good
**Rating:** 6
**Confidence:** 3

**Summary:**

This paper studies the principal component analysis (PCA) in the vertical FL setting, where each party owns a subset of columns in the data matrix. It requires the observations by each client or the server to be differentially private and the error of PCA in the end can be small. The paper proposes an algorithm, where the MPC protocol is utilized and the noise from each party is carefully calculated. In the empirical evaluation, the proposed method is compared with a reasonable baseline and centralized DP algorithm.

**Strengths:**

1. The clarity of this paper is great. The arguments in the paper are well-explained.
2. The problem is well formulated and it is clear the see the advantage of the proposed algorithm over the baseline. The proposed algorithm is independent of $m$ and the baseline highly depends on $m$.
3. The empirical evaluation looks reasonable. The selected dataset covers different range of $(m, n)$.

**Weaknesses:**

1. The utility result (Lemma 3) doesn't show $N$, which is an important factor in vertical FL. It would be great to show how $N$ influences the results empirically.
2. It would be meaningful to present the time/communication cost that happened during the MPC, which is dependent on the choice of $\gamma$.
3. Dataset release in RMGM-OLS [1] would provide another reasonable baseline: unlike the baseline in the paper which adds noise to the data matrix directly, it adds noise after a random projection, which can reasonably reduce the scaling of noise.

[1] Wu, Ruihan, et al. "Differentially Private Multi-Party Data Release for Linear Regression." Uncertainty in Artificial Intelligence. PMLR, 2022

**Questions:**

Please see the "Weakness" above.

---

> ### Author Response · Authors · 2023-11-13
> **Response to Reviewer DSCS**
>
> We would like to thank the reviewer for the thoughtful comments. In what follows, we provide a detailed response. Please kindly let us know if you have any further questions.
>
> ---
>
> **Weakness 1. The utility result of Lemma 3 does not show N, which is an important factor in vertical FL. It would be great to show how N influences the results empirically.**
>
> ---
>
> **Response.** This is an insightful question. The utility result of Lemma 3 and the empirical performance of SPCA is indeed *independent* of $N$, which is exactly why SPCA achieves *comparable performance* as the strong centralized baseline (think of it as $N=1$). To explain, recall that the core idea of SPCA is to aggregate $N$ independent Skellam noises distributed as $Sk(\mu/N)$ from the client side into $Sk(\mu)$, a larger Skellam noise. Here, the scale of $\mu$, which determines the result utility, is in turn determined by the required server-observed level of DP, which is *independent* of $N$.
>
> Instead, what $N$ influences is the level of *client-observed* DP (with the level of server-observed DP fixed), since each client knows her local DP noise out of the overall N contributions. As $N$ increases, the client-observed level of DP becomes stronger since each client’s knowledge about the overall noise becomes smaller. As an illustration, in what follows, we fix the server-observed level of DP to $\epsilon=4$, $\delta=10^-5$ and vary $N$ from {10,20,40,80,160,320,640} for the ACSIncome data with $d=817$.
>
> We report the corresponding levels of *client-observed* $\epsilon$ with $\delta$ fixed to $10^{-5}$ for $\gamma=2^{12}$.
>
> | client count $N$ | 10 | 20 | 40 | 80 | 160 | 320 | 640 |
> | ----------- | ----------- | ----------- | ----------- | ----------- | ----------- | ----------- | ----------- |
> | **client-observed** $\epsilon$ | 9.61 | 8.97 | 8.68 | 8.55 | 8.48 | 8.45| 8.43 |
>
> Note that the change of $\epsilon$ is almost negligible when $N$ reaches $20$. A similar conclusion also applies to other choices of $d$, $\gamma$, as well as different levels of server-observed $\epsilon$. We have revised the draft to incorporate the above discussion.
>
> ---
>
> **Weakness 2. It would be meaningful to present the time/communication cost that happened during the MPC, which is dependent on the choice of $\gamma$.**
>
> ---
>
> **Response.** Thanks for the suggestion. $\gamma$ represents the level of quantization granularity, which is reflected as the number of bits (namely, $\log \gamma$) in communication. For example, quantizing a real number from $0$ to $1$ with $\gamma=2^{12}$ results in a $12$-bit vector. In the case of the classic BGW protocol, the communication is $O(\log \gamma)$ for sharing the secrets. The time complexity, on the other hand,  is independent of $\gamma$ as the computation for constructing the secret shares (matrix multiplication) and reconstructing the secret (polynomial interpolation) are done on the client side, and the algorithms for these computations are usually dependent on the *size* of the input, rather than the *range* of the input. For example, the Newton interpolation on $N$ points ($N$ secret shares of all clients) incurs a time complexity of $O(N^2)$, which is independent of $\gamma$. In general, the time and communication complexity may vary for different MPC protocols and implementations. We have revised the draft to incorporate the above discussion.
>
> ---
>
>
> **Weakness 3. Dataset release in RMGM-OLS [1] would provide another reasonable baseline: unlike the baseline in the paper which adds noise to the data matrix directly, it adds noise after a random projection, which can reasonably reduce the scaling of noise.**
>
> ---
>
> **Response.** Thanks for the reference. Although random projection is an interesting idea, RMGM-OLS by Wu et al. does not apply to our problem. In particular, the utility goal of Wu et al. (see Eq.(1) on page 3 of their paper) considers linear regression, which is a *supervised learning* problem. Based on linear regression, the clients are able to *train* the model weights to better predict the label of the data, a key component of RMGM-OLS. On the contrary, we consider PCA, an *unsupervised learning* problem where there is no target label. As a result, it is not straightforward to apply RMGM-OLS to PCA and compare it with our solution. We have revised the draft to incorporate the above discussion.

---

> > ### Comment · Reviewer_DSCS · 2023-11-23
> > **Official Comment by Reviewer DSCS**
> >
> > Thank the authors for their clarification. It is intuitive that the observed $\varepsilon$ doesn't have too much difference among large $N$, but the difference might be more significant when $N$ is at a small range.

---

> ### Author Response · Authors · 2023-11-19
> **Any further questions or comments?**
>
> We want to thank the reviewer for their comments.
>
> As the deadline for the rebuttal phase is approaching, we would like to encourage the reviewer to engage in the interactive rebuttal to help improve our paper and clarify misunderstandings.

---

### Meta-Review · Area_Chair_GzmU · 2023-12-06

**Metareview:**

There was a large discrepancy in the scores. Hence, I took a closer look at the paper myself. While the paper studies a potentially interesting question, the technical novelty of the paper does not meet the bar for ICLR acceptance. In particular, the main contribution of the paper is to generate the noise needed for privatizing the covariance matrix in a MPC friendly  way. The authors achieve that by generating Skellam noise, which is by-now standard in the literature (https://arxiv.org/pdf/2110.04995.pdf). Given that there is no significant enthusiasm among the reviewers, and the above concern about novelty, we would request the authors to revise their paper in light of the concerns/comments.

**Justification For Why Not Higher Score:**

The novelty of the paper is not high.

**Justification For Why Not Lower Score:**

NA

---

### Decision · Program_Chairs · 2024-01-16

Reject